# Statistical Guarantees for Variational Autoencoders using PAC-Bayesian Theory

**Sokhna Diarra Mbacke**
Université Laval
sokhna-diarra.mbacke.1@ulaval.ca

**Florence Clerc**
McGill University
florence.clerc@mail.mcgill.ca

**Pascal Germain**
Université Laval
pascal.germain@ift.ulaval.ca

## Abstract

Since their inception, Variational Autoencoders (VAEs) have become central in machine learning. Despite their widespread use, numerous questions regarding their theoretical properties remain open. Using PAC-Bayesian theory, this work develops statistical guarantees for VAEs. First, we derive the first PAC-Bayesian bound for posterior distributions conditioned on individual samples from the data-generating distribution. Then, we utilize this result to develop generalization guarantees for the VAE's reconstruction loss, as well as upper bounds on the distance between the input and the regenerated distributions. More importantly, we provide upper bounds on the Wasserstein distance between the input distribution and the distribution defined by the VAE's generative model.

## 1 Introduction

In recent years, deep generative models have exhibited tremendous empirical success. Two of the most important families of generative models are Generative Adversarial Networks (GANs) (Goodfellow et al., 2014) and Variational Autoencoders (Kingma and Welling, 2014; Rezende et al., 2014). GANs take an adversarial approach, whereas VAEs are based on maximum likelihood estimation and variational inference. VAEs comprise two main components: an encoder which parameterizes an approximation of the posterior distribution over the latent variables, and a decoder which parameterizes the likelihood. In addition to generative modelling tasks such as image generation (Vahdat and Kautz, 2020) and text generation (Bowman et al., 2016), VAEs have been successfully applied to other topics such as semi-supervised learning (Kingma et al., 2014), anomaly detection (An and Cho, 2015), and dimensionality reduction (Kaur et al., 2021). However, despite their empirical success, the question of statistical guarantees for the performance of VAEs remains largely open. Namely, how can one certify that VAEs generalize well, both in terms of reconstruction and generation?

PAC-Bayesian theory (McAllester, 1999; Catoni, 2003) is an influential tool of statistical learning theory dedicated to providing generalization bounds for machine learning models. PAC-Bayes has been applied to a wide variety of problems such as classification (Germain et al., 2009; Parrado-Hernández et al., 2012), meta-learning (Amit and Meir, 2018), co-clustering (Seldin and Tishby, 2010), domain adaptation (Germain et al., 2020), and online learning (Haddouche and Guedj, 2022). In recent years, PAC-Bayes has been used to derive non-vacuous generalization bounds for supervised learning algorithms based on neural networks (Dziugaite and Roy, 2018; Pérez-Ortiz et al., 2021). See Guedj (2019) and Alquier (2021) for excellent surveys.

37th Conference on Neural Information Processing Systems (NeurIPS 2023).

The objective of this work is to utilize PAC-Bayesian theory to derive statistical guarantees for VAEs. Our generalization bounds investigate the reconstruction, regeneration, as well as the generation properties of VAEs.

## 1.1 Related Works

In order to explain the empirical success of deep generative models, a lot of attention has been put into deriving theoretical guarantees for these models. Most of the results, however, have been dedicated to GANs and their variants (Arora et al., 2017; Zhang et al., 2018; Liang, 2021; Singh et al., 2018; Schreuder et al., 2021; Biau et al., 2021; Mbacke et al., 2023). A possible explanation for this plethora of theoretical results is the adversarial loss function, which directly offers an estimation of the discrepancy between the input distribution and the generator's distribution. Despite being central tools in modern machine learning, VAEs have not benefited from such a thorough theoretical analysis (Chakrabarty and Das, 2021).

The work of Chakrabarty and Das (2021) studies the regeneration properties of Wasserstein autoencoders (WAEs) (Tolstikhin et al., 2018), which come from the same family as VAEs. Using VC theory, Chakrabarty and Das (2021) derive rates of convergence for the Wasserstein distance between the input distribution and the distribution regenerated by the WAE, as well as the total variation distance between the empirical latent distribution and the latent prior. Taking a more empirical approach, Chérief-Abdellatif et al. (2022) use PAC-Bayes to study the generalization properties of stochastic reconstruction models. They define a $[0, 1]$-bounded reconstruction loss function, then utilize McAllester's bound (McAllester, 2003) to formulate a generalization bound for models with probabilistic neural networks (Langford and Caruana, 2001). Then, they re-scale their loss and compare the empirical results to the reconstruction of standard VAEs on benchmark datasets.

We also mention the work of Mbacke et al. (2023), who developed PAC-Bayesian bounds for the analysis of adversarial generative models. Using McDiarmid's inequality, they proved upper bounds on the distance between the input distribution and the generator's distribution, for WGANs (Arjovsky et al., 2017) and EBGANs (Zhao et al., 2017).

## 1.2 Our Contributions

In this work, we derive theoretical guarantees for variational autoencoders using PAC-Bayesian theory. We provide three types of guarantees: reconstruction guarantees showing that VAEs can successfully reconstruct unseen samples from the input distribution; regeneration guarantees proving upper bounds on the Wasserstein distance between the input distribution and the distribution regenerated by the VAE, given the training set as input; and finally, generation guarantees showing upper bounds on the Wasserstein distance between the data-generating distribution and the VAE's generated distribution defined by the latent prior and the decoder. To the best of our knowledge, these are the first generalization bounds for the standard VAE's reconstruction and regeneration properties, as well as the first statistical guarantees for the VAE's generative model.

In our analysis, the PAC-Bayesian posterior coincides with the variational posterior, which requires the PAC-Bayesian posterior to be conditional. Since, to the best of our knowledge, such PAC-Bayes bounds do not exist in the literature, we start by developing the first PAC-Bayesian bound for conditional posterior distributions. Then, we provide upper bounds for the VAE's performance under two main assumptions: we start by assuming the instance space is bounded, then we take advantage of the manifold hypothesis. Our bounds are functions of the optimization objective of the VAE, namely, the empirical reconstruction loss, and the empirical KL-loss.

The remainder of this paper is organized as follows. In Section 2, we define some preliminary concepts, then briefly introduce VAEs and PAC-Bayesian theory. Section 3 presents our general PAC-Bayesian theorem for conditional posteriors. Then, in Sections 4 and 5, we present our generalization bounds for the reconstruction loss, and the regeneration and generation guarantees.

## 2 Preliminaries

### 2.1 Definitions and Notations

Given metric spaces $(\mathcal{X}, d)$ and $(\mathcal{Y}, d')$, and a real number $K > 0$, a function $f : \mathcal{X} \to \mathcal{Y}$ is $K$-Lipschitz continuous if for any $\mathbf{x}, \mathbf{y} \in \mathcal{X}$, we have

$$d'(f(\mathbf{x}), f(\mathbf{y})) \leq K d(\mathbf{x}, \mathbf{y}).$$

The smallest $K$ such that this condition is satisfied is called the *Lipschitz norm* or *Lipschitz constant* of $f$ and is denoted $\|f\|_{\text{Lip}}$. Moreover, the set of $K$-Lipschitz continuous functions $f : \mathcal{X} \to \mathcal{Y}$ is denoted $\text{Lip}_K(\mathcal{X}, \mathcal{Y})$ (the underlying metrics will be clear from the context).

Throughout the paper, we use lower case letters $p, q$ to denote both probability distributions and their densities w.r.t. the Lebesgue measure. We may add variables between parentheses to improve readability (e.g. $p(\mathbf{z})$ to emphasize that $p$ is a distribution on the space of variables $\mathbf{z}$, and $q(\mathbf{z}|\mathbf{x})$ to indicate that $q$ is a conditional distribution). The set of probability measures on a space $\mathcal{X}$ is denoted $\mathcal{M}_+^1(\mathcal{X})$. The Kullback–Leibler (KL) divergence between $p, q \in \mathcal{M}_+^1(\mathcal{X})$ is denoted $\text{KL}(p \,\|\, q)$. We omit the absolute continuity condition $p \ll q$ in the statements of the results below, since if it is not satisfied, then one may assume the KL divergence is infinite and the bounds hold trivially.

Integral Probability Metrics (IPMs, see Müller (1997)) are a class of pseudo-metrics defined on the space of probability measures. Given a family $\mathcal{F}$ of real-valued functions defined on $\mathcal{X}$, the IPM defined by $\mathcal{F}$ is denoted $d_{\mathcal{F}}$ and defined as

$$d_{\mathcal{F}}(p, q) = \sup_{f \in \mathcal{F}} \left| \int f \, dp - \int f \, dq \right|, \quad \forall p, q \in \mathcal{M}_+^1(\mathcal{X}). \tag{1}$$

Stemming from the theory of optimal transportation (Villani, 2009), the Wasserstein distances (see Definition A.2) are a class of metrics between probability measures. The Wasserstein distance of order 1, also referred to simply as *the Wasserstein distance*, is the IPM defined by the set $\mathcal{F} = \{f : \mathcal{X} \to \mathbb{R} \text{ s.t. } \|f\|_{\text{Lip}} \leq 1\}$.

Finally, we recall the definition of a *pushforward measure*. Let $p$ be a probability distribution on a space $\mathcal{Z}$ and $g : \mathcal{Z} \to \mathcal{X}$ be a measurable function. The pushforward measure defined by $g$ and $p$ and denoted $g \sharp p$ is a probability distribution on $\mathcal{X}$ defined as $g \sharp p(A) = p(g^{-1}(A))$, for any measurable set $A \subseteq \mathcal{X}$. In other words, sampling $\mathbf{x} \sim g \sharp p$ means sampling $\mathbf{z} \sim p$ first, then setting $\mathbf{x} = g(\mathbf{z})$.

### 2.2 Variational Autoencoders

We consider a Euclidean observation space $\mathcal{X}$, a data-generating distribution $\mu \in \mathcal{M}_+^1(\mathcal{X})$, and a latent space $\mathcal{Z} = \mathbb{R}^{d_{\mathcal{Z}}}$. VAEs comprise two main components: the encoder network whose parameters are denoted $\phi$, and the decoder network whose parameters are denoted $\theta$. For simplicity, we may refer to $\phi$ and $\theta$ as the encoder and decoder respectively. The encoder parameterizes a distribution $q_\phi(\mathbf{z}|\mathbf{x})$ over the latent space $\mathcal{Z}$, which is a variational approximation of the Bayesian posterior $p_\theta(\mathbf{z}|\mathbf{x})$. The likelihood $p_\theta(\mathbf{x}|\mathbf{z})$ is parameterized by the decoder network. In this work, we consider the standard VAE, with a standard Gaussian prior $p(\mathbf{z}) = \mathcal{N}(\mathbf{0}, \mathbf{I})$ on $\mathcal{Z}$ and Gaussian latent distributions $q_\phi(\mathbf{z}|\mathbf{x})$. More precisely, for any $\mathbf{x} \in \mathcal{X}$, the distribution $q_\phi(\mathbf{z}|\mathbf{x})$ is a Gaussian distribution with a diagonal covariance matrix $\mathcal{N}(\mu_\phi(\mathbf{x}), \text{diag}(\sigma_\phi^2(\mathbf{x})))$, where

$$\mu_\phi : \mathcal{X} \to \mathcal{Z} = \mathbb{R}^{d_{\mathcal{Z}}} \quad \text{and} \quad \sigma_\phi : \mathcal{X} \to \mathbb{R}_{\geq 0}^{d_{\mathcal{Z}}}.$$

Note that $\text{diag}(\sigma)$ denotes the diagonal matrix whose main diagonal is the vector $\sigma$. In order to simplify some of the expressions below, it may be useful to express the encoder network as a function

$$Q_\phi : \mathcal{X} \to \mathbb{R}^{2d_{\mathcal{Z}}}, \quad \text{where } Q_\phi(\mathbf{x}) = \begin{bmatrix} \mu_\phi(\mathbf{x}) \\ \sigma_\phi(\mathbf{x}) \end{bmatrix}. \tag{2}$$

We express the decoder as a parametric function $g_\theta : \mathcal{Z} \to \mathcal{X}$. For any $\mathbf{x} \in \mathcal{X}$, upon receiving $\mathbf{z} \sim q_\phi(\mathbf{z}|\mathbf{x})$, the decoder's output $g_\theta(\mathbf{z})$ is a reconstruction of $\mathbf{x}$. Given a training set $S = \{\mathbf{x}_1, \ldots, \mathbf{x}_n\}$, the encoder and decoder networks are jointly trained by minimizing the following objective:

$$\mathcal{L}_{\text{VAE}}(\phi, \theta) = \frac{1}{n} \sum_{i=1}^{n} \left[ \mathbb{E}_{\mathbf{z} \sim q_\phi(\mathbf{z}|\mathbf{x}_i)} [-\log p_\theta(\mathbf{x}_i|\mathbf{z})] + \beta \text{KL}(q_\phi(\mathbf{z}|\mathbf{x}_i) \,\|\, p(\mathbf{z})) \right], \tag{3}$$

where the first part of (3) is the *reconstruction loss* and the second part is the KL-divergence between the latent distributions (associated to the training samples) and the prior over the latent space, weighted by a hyperparameter $\beta > 0$ (Higgins et al., 2017). The reconstruction loss measures the similarity between $\mathbf{x}$ and its reconstruction $g_\theta(\mathbf{z})$, and can be defined in many ways. With a Gaussian likelihood, the reconstruction loss is the squared $L_2$ norm $\|\mathbf{x} - g_\theta(\mathbf{z})\|^2$.

After training, the VAE defines a generative model using the prior $p(\mathbf{z})$ and the decoder $g_\theta$ (Kingma and Welling, 2014). The distribution $g_\theta \sharp p(\mathbf{z}) \in \mathcal{M}_+^1(\mathcal{X})$ allows one to generate new samples by first sampling a latent vector from the prior, then passing it through the decoder. We refer to $g_\theta \sharp p(\mathbf{z})$ as the VAE's generated distribution.

## 2.3 A Brief Introduction to PAC-Bayesian Theory

Dating back to McAllester (1999), PAC-Bayesian theory develops high-probability generalization bounds for machine learning algorithms. In essence, PAC-Bayes frames the output of such an algorithm as a posterior distribution over a class of hypotheses, and provides an upper bound on the discrepancy between a model's empirical risk and its population risk.

PAC-Bayes considers the following concepts: a hypothesis class $\mathcal{H}$, a training set $S = \{\mathbf{x}_1, \ldots, \mathbf{x}_n\}$ iid sampled from an unknown distribution $\mu$ over an instance space $\mathcal{X}$[1], and a real-valued loss function $\ell : \mathcal{H} \times \mathcal{X} \to [0, \infty)$. Moreover, the primary goal of PAC-Bayes is to provide generalization bounds uniformly valid for any posterior $q \in \mathcal{M}_+^1(\mathcal{H})$. These bounds are dependent on the empirical performance of $q$ and its closeness to a chosen *prior distribution* $p \in \mathcal{M}_+^1(\mathcal{H})$, as measured by the KL-divergence. The empirical and true risks of a posterior distribution $q \in \mathcal{M}_+^1(\mathcal{H})$ are defined as

$$\hat{\mathcal{R}}_S(q) = \underset{h \sim q(h)}{\mathbb{E}} \left[ \frac{1}{n} \sum_{i=1}^n \ell(h, \mathbf{x}_i) \right] \quad \text{and} \quad \mathcal{R}(q) = \underset{h \sim q(h)}{\mathbb{E}} \left[ \underset{\mathbf{x} \sim \mu}{\mathbb{E}} \ell(h, \mathbf{x}) \right].$$

As an illustration, consider the following PAC-Bayesian bound for bounded loss functions developed by Catoni (2003).

**Theorem 2.1.** *Given a probability measure $\mu$ on $\mathcal{X}$, a hypothesis class $\mathcal{H}$, a prior distribution $p$ on $\mathcal{H}$, a loss function $\ell : \mathcal{H} \times \mathcal{X} \to [0, 1]$, real numbers $\delta \in (0, 1)$ and $\lambda > 0$, with probability at least $1 - \delta$ over the random draw of $S \sim \mu^{\otimes n}$, the following holds for any posterior $q \in \mathcal{M}_+^1(\mathcal{H})$:*

$$\mathcal{R}(q) \le \hat{\mathcal{R}}_S(q) + \frac{\lambda}{8n} + \frac{\mathrm{KL}(q \,\|\, p) + \log \frac{1}{\delta}}{\lambda}.$$

The connection between PAC-Bayesian theory and Bayesian inference was highlighted by Grünwald (2012) and Germain et al. (2016), who showed that with a proper choice of $\lambda$ and the negative log-likelihood as the loss function $\ell$, the optimal posterior minimizing the right-hand side of Catoni's bound is the Bayesian posterior. Note that although the Bayesian posterior is unique (for a given prior and likelihood), a "PAC-Bayesian posterior" could be, in principle, any distribution over $\mathcal{H}$.

In our PAC-Bayesian analysis of VAEs, we will use the latent space $\mathcal{Z}$ as our hypothesis class, so that the VAE's prior will coincide with the PAC-Bayesian prior and the variational posterior $q_\phi(\mathbf{z}|\mathbf{x})$ will stand for our PAC-Bayesian posterior. An immediate concern with this approach is that the encoder's distributions are conditioned on individual samples $\mathbf{x} \sim \mu$, whereas the usual PAC-Bayesian bounds hold for unconditional posteriors $q(h)$. We address this issue in the next section, by developing a novel PAC-Bayesian bound for posterior distributions $q(\cdot|\mathbf{x})$. This general result will be later utilized to analyze VAEs.

## 3 A General PAC-Bayesian Bound with a Conditional Posterior

In this section, we present our general PAC-Bayesian bound with a conditional posterior distribution. Note that the novelty of this result is not the conditioning on observations, since this can be achieved by exploiting the existing PAC-Bayesian bounds. Indeed, Haddouche and Guedj (2022) utilized the general theorem of Rivasplata et al. (2020) to derive bounds for the online learning framework.

---

[1]In supervised learning, the instance space has the form $\mathcal{X} \times \mathcal{Y}$ where $\mathcal{X}$ is a set of features, and $\mathcal{Y}$ a set of labels. We use a more general formulation to encompass the unsupervised learning setting.

Instead, the contribution of Theorem 3.1 is to predict the behavior of $q(h|\mathbf{x})$, for any (previously unseen) $\mathbf{x} \sim \mu$, when the posterior $q$ was only learned using the training samples $\{\mathbf{x}_1, \ldots, \mathbf{x}_n\}$. To the best of our knowledge, this is the first PAC-Bayesian bound where the posterior distribution is a conditional distribution conditioned on individual elements from the instance space. This bound will require the posterior $q$ and the loss function $\ell$ to satisfy the following technical assumption.

**Assumption 1.** We say that a distribution $q(\cdot|\mathbf{x})$ and a loss function $\ell$ satisfy Assumption 1 with a constant $K > 0$ if there exists a family $\mathcal{E}$ of functions $\mathcal{H} \to \mathbb{R}$ such that the following properties hold.

1. The function $\mathbf{x} \mapsto q(\cdot|\mathbf{x})$ is continuous in the following sense: for any $\mathbf{x}_1, \mathbf{x}_2 \in \mathcal{X}$,

$$d_\mathcal{E}\left(q(h|\mathbf{x}_1), q(h|\mathbf{x}_2)\right) \leq K d(\mathbf{x}_1, \mathbf{x}_2).$$

2. For any $\mathbf{x} \in \mathcal{X}$, the function $\ell(\cdot, \mathbf{x}) : \mathcal{H} \to \mathbb{R}$ is in $\mathcal{E}$:

$$\ell(\cdot, \mathbf{x}) \in \mathcal{E}, \quad \text{for any } \mathbf{x} \in \mathcal{X}.$$

Before stating the general result, let us pause and discuss this assumption. Intuitively, the goal of a generalization bound is to predict the behavior of the posterior distribution $q(h|\mathbf{x})$ on previously unseen examples $\mathbf{x} \sim \mu$. Since the posterior $q(h|\mathbf{x})$ is learned by minimizing the loss function $\ell$ on the training samples $S = \{\mathbf{x}_1, \ldots, \mathbf{x}_n\}$, one may need two things to be true.

First, the mapping $\mathbf{x} \mapsto q(h|\mathbf{x})$ has to be somewhat continuous. This is ensured by the first part of Assumption 1, which states that the posterior $q$ is Lipschitz-continuous[2] with respect to the IPM $d_\mathcal{E}$ and the underlying metric $d$ on $\mathcal{X}$. Indeed, this tells us that if $\mathbf{x}_1$ and $\mathbf{x}_2$ are close w.r.t. the underlying metric on $\mathcal{X}$, then $q(h|\mathbf{x}_1)$ and $q(h|\mathbf{x}_2)$ are close, w.r.t. the IPM $d_\mathcal{E}$.

Second, that continuity has to be "understood" by the loss function $\ell$, which corresponds to the second part of the assumption. It states that the loss function's discriminative power is weaker than the one defined by the IPM $d_\mathcal{E}$. In other words, the discrepancy measure used to measure the similarity between the distributions $q(h|\mathbf{x}_1)$ and $q(h|\mathbf{x}_2)$ needs to be just strong enough to fool the loss function into thinking that the distributions are close to each other. An alternate formulation of Assumption 1 is provided in the supplementary material (Remark F.1).

Finally, we emphasize that Assumption 1 is not as restrictive as it may seem at first. For instance, it is satisfied by a VAE's variational posterior, when the encoder and decoder networks have finite Lipschitz norms and the reconstruction loss is defined with the $L_2$ norm (see Proposition 4.1). We are ready to state our first result.

**Theorem 3.1.** *Let $(\mathcal{X}, d)$ be a metric space. Consider a probability measure $\mu$ on $\mathcal{X}$, a hypothesis class $\mathcal{H}$, a prior distribution $p(h)$ on $\mathcal{H}$, a loss function $\ell : \mathcal{H} \times \mathcal{X} \to \mathbb{R}$, real numbers $\delta \in (0, 1)$ and $\lambda > 0$. With probability at least $1 - \delta$ over the random draw of $S \sim \mu^{\otimes n}$, the following holds for any conditional posterior $q(h|\mathbf{x})$ such that Assumption 1 is satisfied by $q(h|\mathbf{x})$ and $\ell$ with a constant $K > 0$:*

$$\mathop{\mathbb{E}}_{\mathbf{x} \sim \mu} \mathop{\mathbb{E}}_{h \sim q(h|\mathbf{x})} \ell(h, \mathbf{x}) - \frac{1}{n} \sum_{i=1}^{n} \mathop{\mathbb{E}}_{h \sim q(h|\mathbf{x}_i)} \ell(h, \mathbf{x}_i) \leq \frac{1}{\lambda} \left[ \sum_{i=1}^{n} \mathrm{KL}(q(h|\mathbf{x}_i) \,||\, p(h)) + \frac{\lambda K}{n} \sum_{i=1}^{n} \mathop{\mathbb{E}}_{\mathbf{x} \sim \mu} d(\mathbf{x}, \mathbf{x}_i) + \right.$$
$$\left. \log \frac{1}{\delta} + n \log \mathop{\mathbb{E}}_{h \sim p(h)} \mathop{\mathbb{E}}_{\mathbf{x} \sim \mu} e^{\frac{\lambda}{n}\left(\mathbb{E}_{\mathbf{x}' \sim \mu} \ell(h, \mathbf{x}') - \ell(h, \mathbf{x})\right)} \right].$$

In order to prove Theorem 3.1, we start by deriving a bound where the expected loss for samples $\mathbf{x} \sim \mu$ is computed w.r.t. distributions $q(h|\mathbf{x}_i)$ associated to the training samples (see Lemma B.1). This result uses standard PAC-Bayesian techniques, with a key difference: we start with $n$ iid hypotheses from the prior $p(h)$, then we perform the change of measure with $n$ posteriors $q_\phi(\mathbf{z}|\mathbf{x}_1), \ldots, q_\phi(\mathbf{z}|\mathbf{x}_n)$, and show that the resulting exponential moment is equal to the one in Theorem 3.1. Moreover, one of the original aspects of this work comes from Assumption 1, which enables us to obtain a bound where the expected loss for $\mathbf{x} \sim \mu$ is computed w.r.t. the posterior $q(h|\mathbf{x})$, associated to $\mathbf{x}$ itself instead of all the training samples. However, the price to pay for having a posterior $q(h|\mathbf{x})$ for each $\mathbf{x} \in \mathcal{X}$ is that the bound depends on $\frac{1}{n} \sum_{i=1}^{n} \mathbb{E}_{\mathbf{x} \sim \mu} d(\mathbf{x}, \mathbf{x}_i)$, which we refer to as the *average distance*.

---

[2] $d_\mathcal{E}$ is a pseudo-metric in the general case, so we abuse the definition by calling this Lipschitz continuity, since the latter concept is only defined for metric spaces.

Applied to supervised learning, Theorem 3.1 bounds the expected risk of a Gibbs posterior $q$ which, upon receiving a previously unseen datapoint $\mathbf{x} \sim \mu$, samples a predictor $h$ *dependent* on $\mathbf{x}$, and uses it to make a prediction. Note that the family $\mathcal{E}$ from Assumption 1 does not appear in the bound, which has nice consequences in practice. Indeed one may pick a loss function $\ell$ that fits the problem, and then find a family $\mathcal{E}$ for which the continuity assumption is satisfied with constant $K$ that is as small as possible.

Note also that, in the tradition of PAC-Bayesian bounds, Theorem 3.1 does not make any assumptions on the nature of the elements of $\mathcal{H}$ (e.g. $\mathcal{H}$ could be a class of functions, a set of neural network's parameters, etc). Therefore, the theorem is very general and could be applied to different domains and models. In the following sections, we will use a specific kind of hypothesis class $\mathcal{H} = \mathcal{Z}$, in order to capture the VAE's latent space.

## 4    Generalization bounds for the Reconstruction Loss

For the remainder of this work, $\|\cdot\|$ denotes the $L_2$ norm, and we assume the instance space $\mathcal{X}$ is Euclidean, and the latent space $\mathcal{Z} = \mathbb{R}^{d_{\mathcal{Z}}}$, where $d_{\mathcal{Z}} > 0$. Both $\mathcal{X}$ and $\mathcal{Z}$ are equipped with the Euclidean distance as the underlying metric. Therefore, if $\mathbf{x}, \mathbf{x}' \in \mathcal{X}$, $d(\mathbf{x}, \mathbf{x}') = \|\mathbf{x} - \mathbf{x}'\|$.

The following assumption states that the encoder and decoder networks have finite Lipschitz norms.

**Assumption 2.** The encoder and decoder are Lipschitz-continuous w.r.t. their inputs, meaning there exist real numbers $K_\phi, K_\theta > 0$ such that for any $\mathbf{x}_1, \mathbf{x}_2 \in \mathcal{X}$ and $\mathbf{z}_1, \mathbf{z}_2 \in \mathcal{Z}$,

$$\|Q_\phi(\mathbf{x}_1) - Q_\phi(\mathbf{x}_2)\| \leq K_\phi \|\mathbf{x}_1 - \mathbf{x}_2\| \tag{4}$$

and

$$\|g_\theta(\mathbf{z}_1) - g_\theta(\mathbf{z}_2)\| \leq K_\theta \|\mathbf{z}_1 - \mathbf{z}_2\|. \tag{5}$$

Recall the definition of $Q_\phi$ from Equation (2). Note that in practice, one can estimate the Lipschitz constant of trained networks (Fazlyab et al., 2019; Latorre et al., 2020) or train the VAE with preset Lipschitz constants (Barrett et al., 2022).

Moreover, we define the reconstruction loss $\ell_{\text{rec}}^\theta$ with the $L_2$ norm, instead of the squared $L_2$ norm, which enables us to exploit the properties of a metric. We discuss this choice in Section 6. In order to be consistent with the PAC-Bayesian framework, we define the loss function as follows: $\ell_{\text{rec}}^\theta : \mathcal{Z} \times \mathcal{X} \to [0, \infty)$,

$$\ell_{\text{rec}}^\theta(\mathbf{z}, \mathbf{x}) = \|\mathbf{x} - g_\theta(\mathbf{z})\|. \tag{6}$$

Our goal is to apply the general bound of Theorem 3.1 to the VAE model. But first, since Theorem 3.1 requires Assumption 1 to be satisfied, we start by showing that if the encoder and decoder networks have finite Lipschitz norms, then Assumption 1 holds.

**Proposition 4.1.** *Consider a VAE with parameters $\phi$ and $\theta$ and let $K_\phi, K_\theta \in \mathbb{R}$ be the Lipschitz norms of the encoder and decoder respectively. Then the variational distribution $q_\phi(\mathbf{z}|\mathbf{x})$ satisfies Assumption 1, with $\mathcal{E} = \{f : \mathcal{Z} \to \mathbb{R} \text{ s.t. } \|f\|_{Lip} \leq K_\theta\}$, $\ell = \ell_{rec}^\theta$, and $K = K_\phi K_\theta$.*

*Proof idea.* The proof of Proposition 4.1 is in Appendix C, we provide a brief summary here. To prove the first part of Assumption 1, we first notice that if $\mathcal{E}$ is the set of real-valued $K_\theta$-Lipschitz continuous functions, then $d_\mathcal{E}$ is a scaling of the Wasserstein distance. In addition, since $W_1 \leq W_2$, using the closed form of the Wasserstein-2 distance between Gaussian distributions, one can show that $d_\mathcal{E}(q_\phi(\cdot|\mathbf{x}_1), q_\phi(\cdot|\mathbf{x}_2)) \leq K_\phi K_\theta \|\mathbf{x}_1 - \mathbf{x}_2\|$. Finally, the second part of the assumption is a consequence of the definition of the loss function and the Lipschitz continuity of the decoder. $\qquad \square$

Proposition 4.1 tells us that Assumption 1 holds for VAEs. Consequently, we can utilize our general bound of Theorem 3.1 to obtain generalization guarantees. This leads to the following general PAC-Bayesian bound for the VAE's reconstruction loss.

**Theorem 4.2.** *Let $\mathcal{X}$ be the instance space, $\mu \in \mathcal{M}_+^1(\mathcal{X})$ the data-generating distribution, $\mathcal{Z}$ the latent space, $p(\mathbf{z}) \in \mathcal{M}_+^1(\mathcal{Z})$ the prior distribution on the latent space, $\theta$ the decoder's parameters,*

$\delta \in (0, 1), \lambda > 0$ *be real numbers. With probability at least* $1 - \delta$ *over the random draw of* $S \sim \mu^{\otimes n}$, *the following holds for any posterior* $q_\phi(\mathbf{z}|\mathbf{x})$:

$$
\mathop{\mathbb{E}}_{\mathbf{x} \sim \mu} \mathop{\mathbb{E}}_{q_\phi(\mathbf{z}|\mathbf{x})} \ell_{rec}^\theta(\mathbf{z}, \mathbf{x}) \leq \frac{1}{n} \sum_{i=1}^n \mathop{\mathbb{E}}_{q_\phi(\mathbf{z}|\mathbf{x}_i)} \ell_{rec}^\theta(\mathbf{z}, \mathbf{x}_i) + \frac{1}{\lambda} \left[ \sum_{i=1}^n \mathrm{KL}(q_\phi(\mathbf{z}|\mathbf{x}_i) \,||\, p(\mathbf{z})) + \right.
$$

$$
\left. \frac{\lambda K_\phi K_\theta}{n} \sum_{i=1}^n \mathop{\mathbb{E}}_{\mathbf{x} \sim \mu} d(\mathbf{x}, \mathbf{x}_i) + \log \frac{1}{\delta} + n \log \mathop{\mathbb{E}}_{\mathbf{z} \sim p(\mathbf{z})} \mathop{\mathbb{E}}_{\mathbf{x} \sim \mu} e^{\frac{\lambda}{n} \left( \mathbb{E}_{\mathbf{x}' \sim \mu} \ell_{rec}^\theta(\mathbf{z}, \mathbf{x}') - \ell_{rec}^\theta(\mathbf{z}, \mathbf{x}) \right)} \right],
$$

*where* $K_\phi$ *and* $K_\theta$ *are the Lipschitz norms of the encoder and the decoder (see* (4) *and* (5)*) and* $\mathbb{E}_{q_\phi(\mathbf{z}|\mathbf{x})}$ *is a shorthand for* $\mathbb{E}_{\mathbf{z} \sim q_\phi(\mathbf{z}|\mathbf{x})}$ .

Note that the choice of the hyperparameter $\beta$ in the VAE's optimization objective (3) correlates with the choice of the hyperparameter $\lambda$ in Theorem 4.2 (e.g. $\lambda = n$ corresponds to $\beta = 1$). Note also that the encoder and decoder are not treated the same way in Theorem 4.2. Indeed, the inequality holds for a given decoder, but uniformly for any encoder. We discuss this subtle difference and its practical consequences in Section 6.

Theorem 4.2 can be seen as a general framework. In order to obtain a useful upper bound, one needs to bound the average distance and the exponential moment on the right-hand side. In the sections below, we provide upper bounds for these terms under various assumptions on the instance space.

## 4.1 Reconstruction Guarantees for Bounded Instance Spaces

In the following theorem, we provide a special case of Theorem 4.2 when the instance space's diameter $\Delta \overset{\text{def}}{=} \sup_{\mathbf{x}, \mathbf{x}' \in \mathcal{X}} d(\mathbf{x}, \mathbf{x}')$ is finite (see Section C.2 for the proof).

**Theorem 4.3.** *Let* $\mathcal{X}$ *be the instance space,* $\Delta < \infty$ *its diameter,* $\mu \in \mathcal{M}_+^1(\mathcal{X})$ *the data-generating distribution,* $\mathcal{Z}$ *the latent space,* $p(\mathbf{z}) \in \mathcal{M}_+^1(\mathcal{Z})$ *the prior on the latent space,* $\theta$ *the decoder's parameters,* $\delta \in (0, 1), \lambda > 0$ *be real numbers. With probability at least* $1 - \delta$ *over the random draw of* $S \sim \mu^{\otimes n}$, *the following holds for any posterior* $q_\phi(\mathbf{z}|\mathbf{x})$:

$$
\mathop{\mathbb{E}}_{\mathbf{x} \sim \mu} \mathop{\mathbb{E}}_{q_\phi(\mathbf{z}|\mathbf{x})} \ell_{rec}^\theta(\mathbf{z}, \mathbf{x}) \leq \frac{1}{n} \sum_{i=1}^n \left\{ \mathop{\mathbb{E}}_{q_\phi(\mathbf{z}|\mathbf{x}_i)} \ell_{rec}^\theta(\mathbf{z}, \mathbf{x}_i) \right\} + \frac{1}{\lambda} \left( \sum_{i=1}^n \mathrm{KL}(q_\phi(\mathbf{z}|\mathbf{x}_i) \,||\, p(\mathbf{z})) + \right.
$$

$$
\left. \lambda K_\phi K_\theta \Delta + \log \frac{1}{\delta} + \frac{\lambda^2 \Delta^2}{8n} \right).
$$

The left-hand side of this inequality is the expected reconstruction loss for samples $\mathbf{x} \sim \mu$, while the right-hand side is the empirical reconstruction and KL losses, plus an additional term depending on the Lipschitz constants of the VAE and the model's diameter.

Note that for real-life datasets, the diameter of the instance space might be very large and non-representative of the structure and complexity of the data. Indeed, it is common to scale image datasets in order to utilize a specific architecture (Radford et al., 2016). In the following section, we provide a special case of Theorem 4.2 under the manifold hypothesis on the data-generating process.

## 4.2 Reconstruction Guarantees under the Manifold Assumption

The manifold assumption (Fodor, 2002; Narayanan and Mitter, 2010; Fefferman et al., 2016) states that most high-dimensional datasets encountered in practice lie close to low-dimensional manifolds. This assumption is exploited by latent variable generative models such as GANs and VAEs, which approximate high-dimensional datasets using transformations of distributions on a low-dimensional space. The works of Schreuder et al. (2021) and Mbacke et al. (2023) provide generalization bounds for GANs, by assuming that the data-generating distribution is a smooth transformation of the uniform distribution on $[0, 1]^{d^*}$, where $d^*$ is the intrinsic dimension. However, since the standard VAE calls for a standard Gaussian prior, in the following theorem, we assume $\mu$ is a smooth transformation of the standard Gaussian distribution $p^*$ on $\mathbb{R}^{d^*}$. We consider the case when $p^*$ is the uniform distribution on $[0, 1]^{d^*}$ in the supplementary material.

**Theorem 4.4.** *Let $\mathcal{X}$ be the instance space, $\mu \in \mathcal{M}_+^1(\mathcal{X})$ the data-generating distribution, $\mathcal{Z}$ the latent space, $p(\mathbf{z}) \in \mathcal{M}_+^1(\mathcal{Z})$ the prior distribution on the latent space, $\theta$ the decoder's parameters, $\delta \in (0,1), \lambda > 0, a > 0$ real numbers. Assume the data-generating distribution $\mu = g^*\sharp p^*$, where $p^*$ is the standard Gaussian distribution on $\mathbb{R}^{d^*}$ and $g^* \in \text{Lip}_{K_*}(\mathbb{R}^{d^*}, \mathcal{X})$. With probability at least $1 - \delta - \frac{nd^*}{2}e^{-a^2/2}$ over the random draw of $S \sim \mu^{\otimes n}$, the following holds for any posterior $q_\phi(\mathbf{z}|\mathbf{x})$:*

$$\mathbb{E}_{\mathbf{x} \sim \mu} \mathbb{E}_{q_\phi(\mathbf{z}|\mathbf{x})} \ell_{rec}^\theta(\mathbf{z}, \mathbf{x}) \le \frac{1}{n} \sum_{i=1}^n \left\{ \mathbb{E}_{q_\phi(\mathbf{z}|\mathbf{x}_i)} \ell_{rec}^\theta(\mathbf{z}, \mathbf{x}_i) \right\} + \frac{1}{\lambda} \left( \sum_{i=1}^n \text{KL}(q_\phi(\mathbf{z}|\mathbf{x}_i) \,\|\, p(\mathbf{z})) + \right.$$
$$\left. \lambda K_\phi K_\theta K_* \sqrt{(1+a^2)d^*} + \log \frac{1}{\delta} + \frac{\lambda^2 K_*^2}{2n} \right).$$

Let us clarify the role of the new parameter $a > 0$. Each training sample $\mathbf{x}_i \in S$ can be expressed as $\mathbf{x}_i = g^*(\mathbf{w}_i)$, where $\mathbf{w}_i \sim p^*$. Since $p^*$ is the standard Gaussian distribution on $\mathbb{R}^{d^*}$, all samples $\mathbf{w}_i$ will be inside a hypercube $[-a, a]^{d^*}$, with high probability. This uncertainty is reflected in the lowered confidence (from $1 - \delta$ in Theorem 4.2 to $1 - \delta - \frac{nd^*}{2}e^{-a^2/2}$ in Theorem 4.4), and can be controlled by choosing a large enough value of $a$. The proof of Theorem 4.4 is in the supplementary material (Section C.3), we provide a short summary below.

*Proof idea.* The proof starts with Theorem 4.2, and uses the assumptions of Theorem 4.4 to obtain upper bounds on the exponential moment and the average distance. To derive the upper bound on the exponential moment, we observe that the function $\mathbf{z} \mapsto \ell_{rec}^\theta(\mathbf{z}, \mathbf{x})$ is $K_*$-Lipschitz continuous, then we use a dimension-free upper bound on the MGF of Lipschitz-continuous functions of Gaussian random variables. Furthermore, we obtain the upper bound on the average distance $\frac{1}{n} \sum_{i=1}^n \mathbb{E}_{\mathbf{x} \sim \mu} \|\mathbf{x} - \mathbf{x}_i\|$, by using Holder's inequality and the expectation of a non-central $\chi^2$ distribution. Then, we upper-bound the probability that $\mathbf{w}_i \in [-a, a]^{d^*}$ for all $1 \le i \le n$ using the error function and Bernoulli's inequality. Finally, we use the union bound to update the overall confidence. $\qquad \square$

## 5 Generalization Bounds for Regeneration and Generation

Let $\hat{\mu}_{\phi,\theta}$ be the *empirical regenerated distribution*, meaning

$$\hat{\mu}_{\phi,\theta} = \frac{1}{n} \sum_{i=1}^n g_\theta \sharp q_\phi(\mathbf{z}|\mathbf{x}_i). \tag{7}$$

In other words, sampling $\mathbf{x} \sim \hat{\mu}_{\phi,\theta}$ is done by sampling $\mathbf{z} \sim q_\phi(\mathbf{z}|\mathbf{x}_i)$ where $i$ is uniformly sampled from $\{1, \dots, n\}$, then passing $\mathbf{z}$ through the decoder: $\mathbf{x} = g_\theta(\mathbf{z})$. It is therefore the distribution regenerated by the VAE, given the training set $S = \{\mathbf{x}_1, \dots, \mathbf{x}_n\}$ as input.

In this section, we provide statistical guarantees on the regenerative and generative properties of VAEs. More precisely, we derive upper bounds for the quantities $W_1(\mu, \hat{\mu}_{\phi,\theta})$ and $W_1(\mu, g_\theta \sharp p(\mathbf{z}))$. Note that the average distance term does not appear in the bounds of this section. This is because instead of relying on Theorem 3.1, the results of this section depend upon a preliminary lemma (Lemma B.1), which does not necessitate Assumption 1.

### 5.1 Regeneration and Generation Guarantees for Bounded Instance Spaces

The following theorem presents our first upper bound on the distance between the input distribution and the empirical regenerated distribution.

**Theorem 5.1.** *Under the definitions and assumptions of Theorem 4.3, we have that with probability at least $1 - \delta$ over the random draw of $S \sim \mu^{\otimes n}$, the following holds for any posterior $q_\phi(\mathbf{z}|\mathbf{x})$:*

$$W_1(\mu, \hat{\mu}_{\phi,\theta}) \le \frac{1}{n} \sum_{i=1}^n \left\{ \mathbb{E}_{q_\phi(\mathbf{z}|\mathbf{x}_i)} \ell_{rec}^\theta(\mathbf{z}, \mathbf{x}_i) \right\} + \frac{1}{\lambda} \left( \sum_{i=1}^n \text{KL}(q_\phi(\mathbf{z}|\mathbf{x}_i) \,\|\, p(\mathbf{z})) + \log \frac{1}{\delta} + \frac{\lambda^2 \Delta^2}{8n} \right).$$

As we can see, the right-hand side of Theorem 5.1 depends on the empirical reconstruction loss and KL-divergence. This guarantees that as the VAE's empirical risk decreases, the regenerated distribution gets closer to the data-generating distribution. The proof of Theorem 5.1 exploits the fact

that the underlying metric on $\mathcal{X}$ is the Euclidean distance $d(\mathbf{x}, \mathbf{x}') = \|\mathbf{x} - \mathbf{x}'\|$, which is also used to define the reconstruction loss $\ell_{rec}^\theta$ (see Equation 6). The full proof can be found in Appendix D.

The following theorem provides an upper bound of the distance between the input distribution and the VAE's generated distribution.

**Theorem 5.2.** *Under the definitions and assumptions of Theorem 4.3, we have that with probability at least $1 - \delta$ over the random draw of $S \sim \mu^{\otimes n}$, the following holds for any posterior $q_\phi(\mathbf{z}|\mathbf{x})$:*

$$W_1(\mu, g_\theta \sharp p(\mathbf{z})) \leq \frac{1}{n} \sum_{i=1}^n \left\{ \mathop{\mathbb{E}}_{q_\phi(\mathbf{z}|\mathbf{x}_i)} \ell_{rec}^\theta(\mathbf{z}, \mathbf{x}_i) \right\} + \frac{1}{\lambda} \left( \sum_{i=1}^n \mathrm{KL}(q_\phi(\mathbf{z}|\mathbf{x}_i) \,\|\, p(\mathbf{z})) + \right.$$
$$\left. \log \frac{1}{\delta} + \frac{\lambda^2 \Delta^2}{8n} \right) + \frac{K_\theta}{n} \sum_{i=1}^n \sqrt{\|\mu_\phi(\mathbf{x}_i)\|^2 + \left\|\sigma_\phi(\mathbf{x}_i) - \vec{1}\right\|^2},$$

*where $\vec{1} \in \mathbb{R}^{d_\mathcal{Z}}$ denotes the vector whose entries are all $1$.*

The right-hand side of Theorem 5.2 is equal to the right-hand side of Theorem 5.1, plus an additional term depending on the Wasserstein-2 distance $W_2(q_\phi(\mathbf{z}|\mathbf{x}_i), p(\mathbf{z}))$, which is used in the proof because of its closed form for Gaussian distributions. Hence, the right-hand side of Theorem 5.2 augments the VAE's optimization objective with $W_2(q_\phi(\mathbf{z}|\mathbf{x}_i), p(\mathbf{z}))$, suggesting that a good generative performance may require the latent codes to be even closer to the prior. This is consistent with the findings of Zhao et al. (2019), who showed that in order to improve generative performance, the latent codes need to be much closer to the prior, which may disrupt the balance between reconstruction loss and KL-loss.

## 5.2 Regeneration and Generation Guarantees under the Manifold Assumption

Similar to what we did in Section 4.2, we assume that the data-generating distribution is a smooth transformation of the standard Gaussian distribution on $\mathbb{R}^{d^*}$, where $d^*$ is the intrinsic dimension of the dataset. This yields the following results.

**Theorem 5.3.** *Under the definitions and assumptions of Theorem 4.4, with probability at least $1 - \delta$ over the random draw of $S \sim \mu^{\otimes n}$, the following holds for any posterior $q_\phi(\mathbf{z}|\mathbf{x})$:*

$$W_1(\mu, \hat{\mu}_{\phi,\theta}) \leq \frac{1}{n} \sum_{i=1}^n \left\{ \mathop{\mathbb{E}}_{q_\phi(\mathbf{z}|\mathbf{x}_i)} \ell_{rec}^\theta(\mathbf{z}, \mathbf{x}_i) \right\} + \frac{1}{\lambda} \left( \sum_{i=1}^n \mathrm{KL}(q_\phi(\mathbf{z}|\mathbf{x}_i) \,\|\, p(\mathbf{z})) + \log \frac{1}{\delta} + \frac{\lambda^2 K_*^2}{2n} \right).$$

Note that the intrinsic and extrinsic dimensions do not explicitly appear in this inequality, although they may affect the reconstruction and KL loss.

We now present our last result, an upper bound on the Wasserstein distance between the input distribution and the VAE's generated distribution, under the manifold assumption.

**Theorem 5.4.** *Under the definitions and assumptions of Theorem 4.4, with probability at least $1 - \delta$ over the random draw of $S \sim \mu^{\otimes n}$, the following holds for any posterior $q_\phi(\mathbf{z}|\mathbf{x})$ :*

$$W_1(\mu, g_\theta \sharp p(\mathbf{z})) \leq \frac{1}{n} \sum_{i=1}^n \left\{ \mathop{\mathbb{E}}_{q_\phi(\mathbf{z}|\mathbf{x}_i)} \ell_{rec}^\theta(\mathbf{z}, \mathbf{x}_i) \right\} + \frac{1}{\lambda} \left( \sum_{i=1}^n \mathrm{KL}(q_\phi(\mathbf{z}|\mathbf{x}_i) \,\|\, p(\mathbf{z})) + \right.$$
$$\left. \log \frac{1}{\delta} + \frac{\lambda^2 K_*^2}{2n} \right) + \frac{K_\theta}{n} \sum_{i=1}^n \sqrt{\|\mu_\phi(\mathbf{x}_i)\|^2 + \left\|\sigma_\phi(\mathbf{x}_i) - \vec{1}\right\|^2},$$

*where $\vec{1} \in \mathbb{R}^{d_\mathcal{Z}}$ denotes the vector whose entries are all $1$.*

Theorem 5.2 and Theorem 5.4 show that by minimizing the VAE's objective, one is also minimizing the Wasserstein distance between the input distribution and the VAE's generated distribution.

From the upper bounds given by Theorems 5.1 and 5.3, one can deduce rates of convergence of $O(n^{-1/2})$ (when $\lambda \approx \sqrt{n}$) for the empirical regenerated distribution. Note that $\lambda \approx n$ leads to the much faster rate of $n^{-1}$, but then the bounds do not converge to the empirical risk, but to a larger positive number, dependent on the input distribution. Similarly, Theorems 5.2 and 5.4 provide rates of convergence of $O(n^{-1/2})$ for the VAE's generated distribution.

# 6 Discussion and Conclusion

**The different treatments of $\theta$ and $\phi$.**  The bounds we've presented in this work hold for a given decoder $\theta$, but uniformly for all encoders. In practice, this means that the risk certificate has to be computed using samples different from the ones used to train the VAE. This is different from the usual PAC-Bayesian trick (Germain et al., 2009; Parrado-Hernández et al., 2012; Pérez-Ortiz et al., 2021, see also Remark F.2) of splitting the training set to learn the prior, then training the model on the whole training set, because the decoder and encoder are jointly optimized. Instead, one has to make sure that the model is only trained on samples distinct from the ones used to compute the bound. The same method would be necessary when computing the risk certificates given by the recent PAC-Bayesian bounds of Rivasplata et al. (2020) and Haddouche and Guedj (2022), since those bounds are not uniformly valid for any posterior.

**The reconstruction loss.**  In our bounds, the reconstruction loss is the $L_2$ norm (RMSE), instead of the squared $L_2$ norm (MSE). In practice, one can still optimize a VAE with the MSE (or any other reconstruction loss, e.g. the cross entropy loss), and then compute the bounds using the RMSE. However, if the reconstruction loss is not the RMSE, then the optima of the chosen optimization objective might differ from the ones minimizing the right-hand side of the bounds. Therefore, if the goal is to minimize the bounds, one should utilize the RMSE as the reconstruction loss.

**Conclusion.**  It is common, when applying PAC-Bayesian theory to new problems, to add additional stochasticity in order to account for the PAC-Bayesian distributions on the hypothesis class. For instance, Mbacke et al. (2023) added distributions on the parameters of a WGAN's generator, in order to perform a PAC-Bayesian analysis. However, because of the seamless integration of the PAC-Bayesian and VAE frameworks, such modification to the original problem has been avoided in this work. We matched the prior and posterior distributions on the VAE's latent space to the PAC-Bayesian prior and posterior, which allowed us to recover the VAE's optimization objective. We provide preliminary experiments on synthetic datasets in the supplementary material.

This work is a humble contribution to the theoretical understanding of VAEs. We developed novel PAC-Bayesian bounds suited to the analysis of VAEs and provided generalizations bounds for the VAE's reconstruction loss. In addition, we also derived upper bounds on the Wasserstein distance between the input distribution and the VAE's generative model's distribution. These bounds depend on the VAE's empirical optimization objective and the data-generating process. By integrating the VAE and PAC-Bayesian frameworks, we hope to establish PAC-Bayesian theory as a prime tool for the theoretical analysis of VAEs.

## Acknowledgements

This research is supported by the Canada CIFAR AI Chair Program, and the NSERC Discovery grant RGPIN-2020- 07223. F. Clerc is funded by IVADO through the DEEL Project CRDPJ 537462 18 and by a grant from NSERC.

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
