# Statistical Guarantees for Variational Autoencoders using PAC-Bayesian Theory: Supplementary Material

**Sokhna Diarra Mbacke**
Université Laval
sokhna-diarra.mbacke.1@ulaval.ca

**Florence Clerc**
McGill University
florence.clerc@mail.mcgill.ca

**Pascal Germain**
Université Laval
pascal.germain@ift.ulaval.ca

## A    Preliminaries

**Definition A.1** (Coupling). Let $p, q \in \mathcal{M}_+^1(\mathcal{X})$. A distribution $\gamma$ on $\mathcal{X} \times \mathcal{X}$ is a coupling (Villani, 2009) of $p$ and $q$ if for every measurable set $B \subset \mathcal{X}$, $\gamma(B \times \mathcal{X}) = p(B)$ and $\gamma(\mathcal{X} \times B) = q(B)$. In other words, a coupling of $p$ and $q$ is a distribution on $\mathcal{X} \times \mathcal{X}$ whose marginals are $p$ and $q$ respectively.

For example, the product measure $p \otimes q$ is a coupling of $p$ and $q$.

**Definition A.2** (Wasserstein distances). Let $(\mathcal{X}, d)$ be a Polish metric space and $p, q \in \mathcal{M}_+^1(\mathcal{X})$ Given a real number $k \geq 1$, the Wasserstein-$k$ distance $W_k$ is defined as

$$W_k(p, q) = \left( \inf_{\pi \in \Gamma(p,q)} \int d(\mathbf{x}, \mathbf{y})^k \, d\pi(\mathbf{x}, \mathbf{y}) \right)^{1/k},$$

where $\Gamma(p, q)$ denotes the set of couplings of $p$ and $q$ (see Definition A.1 above). As stated in the main paper, $W_1$ is referred to as the Wasserstein distance.

Given two Gaussian distributions $p = \mathcal{N}(\mu_1, \Sigma_1)$ and $q = \mathcal{N}(\mu_2, \Sigma_2)$ on $\mathbb{R}^{d^*}$, the Wasserstein-2 distance has the following closed form (Givens and Shortt, 1984):

$$W_2(p, q)^2 = \|\mu_1 - \mu_2\|^2 + \mathrm{Tr}\left( \Sigma_1 + \Sigma_2 - 2 \left( \Sigma_1^{1/2} \Sigma_2 \Sigma_1^{1/2} \right)^{1/2} \right). \tag{A.1}$$

This expression can be greatly simplified when the distributions have diagonal covariance matrices. Indeed, if $\Sigma_1 = \mathrm{diag}(\sigma_1^2)$ and $\Sigma_2 = \mathrm{diag}(\sigma_2^2)$ where $\sigma_1, \sigma_2 \in \mathbb{R}^{d^*}$, then the product of the covariance matrices commutes $\Sigma_1 \Sigma_2 = \Sigma_2 \Sigma_1$ and we get

$$\left( \Sigma_1^{1/2} \Sigma_2 \Sigma_1^{1/2} \right)^{1/2} = \Sigma_1^{1/2} \Sigma_2^{1/2},$$

which, combined with the symmetry of covariance matrices and the definition of the Frobenius norm $\|\cdot\|_{\mathrm{Fr}}$ (Petersen and Pedersen, 2008), implies

$$\mathrm{Tr}\left( \Sigma_1 + \Sigma_2 - 2 \left( \Sigma_1^{1/2} \Sigma_2 \Sigma_1^{1/2} \right)^{1/2} \right) = \left\| \Sigma_1^{1/2} - \Sigma_2^{1/2} \right\|_{\mathrm{Fr}}^2 = \|\sigma_1 - \sigma_2\|^2.$$

Hence, if $p = \mathcal{N}(\mu_1, \mathrm{diag}(\sigma_1^2))$ and $q = \mathcal{N}(\mu_2, \mathrm{diag}(\sigma_2^2))$, then the Wasserstein-2 distance between $p$ and $q$ is

$$W_2(p, q) = \|\mu_1 - \mu_2\|^2 + \|\sigma_1 - \sigma_2\|^2. \tag{A.2}$$

We will use this equality to prove some of the results of Section 5.

The following change of measure theorem dates back to Donsker and Varadhan (1976) and has been used in the proof of many PAC-Bayesian theorems. A proof can be found in Boucheron et al. (2013, Corollary 4.15).

**Proposition A.1** (Donsker-Varadhan change of measure). *Let $p, q$ be probability measures on a space $\mathcal{H}$ such that $q \ll p$, and let $g : \mathcal{H} \to \mathbb{R}$ be a function such that $\mathbb{E}_{h \sim p}\, e^{g(h)} < \infty$. Then,*

$$\mathop{\mathbb{E}}_{h \sim p}\, e^{g(h)} \geq e^{\mathbb{E}_{h \sim q}[g(h)] - \mathrm{KL}(q \,\|\, p)}.$$

There are many different formulations of this proposition, we chose a formulation that facilitates readability of the proof of the following lemma.

# B  Proofs of the results in Section 3

We state and prove our first result. Note that the following lemma does not use Assumption 1. Moreover, the main difference between the inequality of this lemma and the one of Theorem 3.1 is the left-hand side. In Lemma B.1, the expected loss for samples $\mathbf{x} \sim \mu$ is computed w.r.t. distributions $q(h|\mathbf{x}_i)$ associated to the training samples. In contrast, in Theorem 3.1, the expected loss for each $\mathbf{x} \sim \mu$ is computed w.r.t. the distribution $q(h|\mathbf{x})$ associated to $\mathbf{x}$ itself.

**Lemma B.1.** *Let $\mathcal{X}$ be the instance space, $\mu \in \mathcal{M}_+^1(\mathcal{X})$ the data-generating distribution, $\mathcal{H}$ the hypothesis class, $\ell : \mathcal{H} \times \mathcal{X} \to \mathbb{R}$ the loss function, $p(h) \in \mathcal{M}_+^1(\mathcal{H})$ the prior distribution and $\delta \in (0, 1), \lambda > 0$ real numbers. Then with probability at least $1 - \delta$ over the random draw of the training set $S = \{\mathbf{x}_1, \ldots, \mathbf{x}_n\} \sim \mu^{\otimes n}$, the following holds for any conditional posterior $q(h|\mathbf{x}) \in \mathcal{M}_+^1(\mathcal{H})$:*

$$\frac{1}{n} \sum_{i=1}^n \left\{ \mathop{\mathbb{E}}_{h \sim q(h|\mathbf{x}_i)} \mathop{\mathbb{E}}_{\mathbf{x} \sim \mu} \ell(h, \mathbf{x}) \right\} \leq \frac{1}{n} \sum_{i=1}^n \left\{ \mathop{\mathbb{E}}_{h \sim q(h|\mathbf{x}_i)} \ell(h, \mathbf{x}_i) \right\} + \frac{1}{\lambda} \left[ \sum_{i=1}^n \mathrm{KL}(q(h|\mathbf{x}_i) \,\|\, p(h)) + \right.$$
$$\left. \log \frac{1}{\delta} + n \log \mathop{\mathbb{E}}_{\mathbf{x} \sim \mu} \mathop{\mathbb{E}}_{h \sim p(h)} \exp \left[ \frac{\lambda}{n} \left( \mathop{\mathbb{E}}_{\mathbf{x}' \sim \mu} [\ell(h, \mathbf{x}')] - \ell(h, \mathbf{x}) \right) \right] \right].$$
(B.1)

*Proof.* First, we consider a set $H = \{h_1, \ldots, h_n\} \sim p(h)^{\otimes n}$ iid sampled from $p(h)$. By applying Markov's inequality to the positive random variable $Y$, defined as

$$Y \stackrel{\text{def}}{=} \mathop{\mathbb{E}}_{H \sim p(h)^{\otimes n}} \exp \left[ \frac{\lambda}{n} \sum_{i=1}^n \left\{ \mathop{\mathbb{E}}_{\mathbf{x} \sim \mu} [\ell(h_i, \mathbf{x})] - \ell(h_i, \mathbf{x}_i) \right\} \right],$$

we obtain that with probability at least $1 - \delta$ over the draw of $S \sim \mu^{\otimes n}$, $Y \leq \frac{1}{\delta} \mathbb{E}[Y]$, meaning

$$\mathop{\mathbb{E}}_{H \sim p(h)^{\otimes n}} \exp \left[ \frac{\lambda}{n} \sum_{i=1}^n \left\{ \mathop{\mathbb{E}}_{\mathbf{x} \sim \mu} [\ell(h_i, \mathbf{x})] - \ell(h_i, \mathbf{x}_i) \right\} \right] \leq$$
$$\frac{1}{\delta} \mathop{\mathbb{E}}_{S \sim \mu^{\otimes n}} \mathop{\mathbb{E}}_{H \sim p(h)^{\otimes n}} \exp \left[ \frac{\lambda}{n} \sum_{i=1}^n \left\{ \mathop{\mathbb{E}}_{\mathbf{x} \sim \mu} [\ell(h_i, \mathbf{x})] - \ell(h_i, \mathbf{x}_i) \right\} \right].$$
(B.2)

Let us focus on the left-hand side of (B.2). We have

$$
\underset{H \sim p(h)^{\otimes n}}{\mathbb{E}} \exp \left[ \frac{\lambda}{n} \sum_{i=1}^{n} \left\{ \underset{\mathbf{x} \sim \mu}{\mathbb{E}} [\ell(h_i, \mathbf{x})] - \ell(h_i, \mathbf{x}_i) \right\} \right]
$$

$$
= \underset{H \sim p(h)^{\otimes n}}{\mathbb{E}} \prod_{i=1}^{n} \exp \left[ \frac{\lambda}{n} \left( \underset{\mathbf{x} \sim \mu}{\mathbb{E}} [\ell(h_i, \mathbf{x})] - \ell(h_i, \mathbf{x}_i) \right) \right]
$$

$$
= \prod_{i=1}^{n} \underset{h_i \sim p(h)}{\mathbb{E}} \exp \left[ \frac{\lambda}{n} \left( \underset{\mathbf{x} \sim \mu}{\mathbb{E}} [\ell(h_i, \mathbf{x})] - \ell(h_i, \mathbf{x}_i) \right) \right]
$$

$$
= \prod_{i=1}^{n} \underset{h \sim p(h)}{\mathbb{E}} \exp \left[ \frac{\lambda}{n} \left( \underset{\mathbf{x} \sim \mu}{\mathbb{E}} [\ell(h, \mathbf{x})] - \ell(h, \mathbf{x}_i) \right) \right]
$$

$$
\geq \prod_{i=1}^{n} \exp \left[ \underset{h \sim q(h|\mathbf{x}_i)}{\mathbb{E}} \left[ \frac{\lambda}{n} \left( \underset{\mathbf{x} \sim \mu}{\mathbb{E}} [\ell(h, \mathbf{x})] - \ell(h, \mathbf{x}_i) \right) \right] - \mathrm{KL}(q(h|\mathbf{x}_i) \,\|\, p(h)) \right],
$$

where the inequality uses the Donsker-Varadhan change of measure theorem (Proposition A.1). Applying the logarithm, we obtain

$$
\log \underset{H \sim p(h)^{\otimes n}}{\mathbb{E}} \exp \left[ \frac{\lambda}{n} \sum_{i=1}^{n} \left\{ \underset{\mathbf{x} \sim \mu}{\mathbb{E}} [\ell(h_i, \mathbf{x})] - \ell(h_i, \mathbf{x}_i) \right\} \right]
$$

$$
\geq \log \prod_{i=1}^{n} \exp \left[ \underset{h \sim q(h|\mathbf{x}_i)}{\mathbb{E}} \left[ \frac{\lambda}{n} \left( \underset{\mathbf{x} \sim \mu}{\mathbb{E}} [\ell(h, \mathbf{x})] - \ell(h, \mathbf{x}_i) \right) \right] - \mathrm{KL}(q(h|\mathbf{x}_i) \,\|\, p(h)) \right]
$$

$$
= \sum_{i=1}^{n} \left( \underset{h \sim q(h|\mathbf{x}_i)}{\mathbb{E}} \left[ \frac{\lambda}{n} \left( \underset{\mathbf{x} \sim \mu}{\mathbb{E}} [\ell(h, \mathbf{x})] - \ell(h, \mathbf{x}_i) \right) \right] - \mathrm{KL}(q(h|\mathbf{x}_i) \,\|\, p(h)) \right)
$$

$$
= \frac{\lambda}{n} \sum_{i=1}^{n} \underset{h \sim q(h|\mathbf{x}_i)}{\mathbb{E}} \left[ \underset{\mathbf{x} \sim \mu}{\mathbb{E}} [\ell(h, \mathbf{x})] - \ell(h, \mathbf{x}_i) \right] - \sum_{i=1}^{n} \mathrm{KL}(q(h|\mathbf{x}_i) \,\|\, p(h)).
$$

This, combined with (B.2) yields

$$
\frac{\lambda}{n} \sum_{i=1}^{n} \underset{h \sim q(h|\mathbf{x}_i)}{\mathbb{E}} \left[ \underset{\mathbf{x} \sim \mu}{\mathbb{E}} [\ell(h, \mathbf{x})] - \ell(h, \mathbf{x}_i) \right] - \sum_{i=1}^{n} \mathrm{KL}(q(h|\mathbf{x}_i) \,\|\, p(h)) \leq
$$
$$
\log \frac{1}{\delta} \underset{S \sim \mu^{\otimes n}}{\mathbb{E}} \underset{H \sim p(h)^{\otimes n}}{\mathbb{E}} \exp \left[ \frac{\lambda}{n} \sum_{i=1}^{n} \left\{ \underset{\mathbf{x} \sim \mu}{\mathbb{E}} [\ell(h_i, \mathbf{x})] - \ell(h_i, \mathbf{x}_i) \right\} \right].
$$
(B.3)

It remains to show that the exponential moment on the right-hand side of Equation (B.3) can be modified by replacing the expectation w.r.t. $p(h)^{\otimes n}$ with an expectation w.r.t. $p(h)$. Similar to what

we did in the first part of the first derivation, we can use Fubini's theorem to obtain

$$\mathbb{E}_{S \sim \mu^{\otimes n}} \mathbb{E}_{H \sim p(h)^{\otimes n}} \exp \left[ \frac{\lambda}{n} \sum_{i=1}^{n} \left\{ \mathbb{E}_{\mathbf{x} \sim \mu} [\ell(h_i, \mathbf{x})] - \ell(h_i, \mathbf{x}_i) \right\} \right]$$

$$= \mathbb{E}_{S \sim \mu^{\otimes n}} \prod_{i=1}^{n} \mathbb{E}_{h \sim p(h)} \exp \left[ \frac{\lambda}{n} \left( \mathbb{E}_{\mathbf{x} \sim \mu} [\ell(h, \mathbf{x})] - \ell(h, \mathbf{x}_i) \right) \right]$$

$$= \prod_{i=1}^{n} \mathbb{E}_{\mathbf{x}_i \sim \mu} \mathbb{E}_{h \sim p(h)} \exp \left[ \frac{\lambda}{n} \left( \mathbb{E}_{\mathbf{x} \sim \mu} [\ell(h, \mathbf{x})] - \ell(h, \mathbf{x}_i) \right) \right]$$

$$= \prod_{i=1}^{n} \mathbb{E}_{\mathbf{x} \sim \mu} \mathbb{E}_{h \sim p(h)} \exp \left[ \frac{\lambda}{n} \left( \mathbb{E}_{\mathbf{x}' \sim \mu} [\ell(h, \mathbf{x}')] - \ell(h, \mathbf{x}) \right) \right]$$

$$= \left( \mathbb{E}_{\mathbf{x} \sim \mu} \mathbb{E}_{h \sim p(h)} \exp \left[ \frac{\lambda}{n} \left( \mathbb{E}_{\mathbf{x}' \sim \mu} [\ell(h, \mathbf{x}')] - \ell(h, \mathbf{x}) \right) \right] \right)^n.$$

Hence,

$$\log \mathbb{E}_{S \sim \mu^{\otimes n}} \mathbb{E}_{H \sim p(h)^{\otimes n}} \exp \left[ \frac{\lambda}{n} \sum_{i=1}^{n} \left\{ \mathbb{E}_{\mathbf{x} \sim \mu} [\ell(h_i, \mathbf{x})] - \ell(h_i, \mathbf{x}_i) \right\} \right] =$$

$$n \log \mathbb{E}_{\mathbf{x} \sim \mu} \mathbb{E}_{h \sim p(h)} \exp \left[ \frac{\lambda}{n} \left( \mathbb{E}_{\mathbf{x}' \sim \mu} [\ell(h, \mathbf{x}')] - \ell(h, \mathbf{x}) \right) \right].$$

Combining this equation with Equation (B.3) yields the theorem. □

The reader familiar with PAC-Bayes bounds may notice that the proof of Lemma B.1 is similar to the usual derivation of PAC-Bayesian bounds, with a key difference. We start with an iid set of $n$ hypotheses sampled from the prior, which allows us to apply the change of measure theorem to $n$ posteriors $q(h|\mathbf{x}_1), \dots, q(h|\mathbf{x}_n)$. Then, we show that the exponential moment obtained with $n$ hypotheses instead of one is equal to the exponential moment obtained with one hypothesis.

## B.1 Proof of Theorem 3.1

The first summand on the left-hand side of Lemma B.1 is the risk on samples $\mathbf{x} \sim \mu$, when the hypotheses are uniformly sampled from $q(h|\mathbf{x}_i), 1 \leq i \leq n$. In order to replace $q(h|\mathbf{x}_i)$ by $q(h|\mathbf{x})$ in that term and derive Theorem 3.1, we utilize Assumption 1.

First, recall that Theorem 3.1 states that under the assumptions of Lemma B.1, if Assumption 1 holds with a constant $K > 0$, then the following inequality holds with probability at least $1 - \delta$:

$$\mathbb{E}_{\mathbf{x} \sim \mu} \mathbb{E}_{h \sim q(h|\mathbf{x})} \ell(h, \mathbf{x}) - \frac{1}{n} \sum_{i=1}^{n} \mathbb{E}_{h \sim q(h|\mathbf{x}_i)} \ell(h, \mathbf{x}_i) \leq \frac{1}{\lambda} \left[ \sum_{i=1}^{n} \mathrm{KL}(q(h|\mathbf{x}_i) \,\|\, p(h)) + \frac{\lambda K}{n} \sum_{i=1}^{n} \mathbb{E}_{\mathbf{x} \sim \mu} [d(\mathbf{x}, \mathbf{x}_i)] + \right.$$

$$\left. \log \frac{1}{\delta} + n \log \mathbb{E}_{\mathbf{x} \sim \mu} \mathbb{E}_{h \sim p(h)} e^{\frac{\lambda}{n} \left( \mathbb{E}_{\mathbf{x}' \sim \mu} [\ell(h, \mathbf{x}')] - \ell(h, \mathbf{x}) \right)} \right].$$

(B.4)

*Proof of Theorem 3.1.* Using the definition of an IPM and Assumption 1, for any $\mathbf{x}_i \in S, \mathbf{x} \in \mathcal{X}$, we have

$$\mathbb{E}_{h \sim q(h|\mathbf{x})} \ell(h, \mathbf{x}) - \mathbb{E}_{h \sim q(h|\mathbf{x}_i)} \ell(h, \mathbf{x}) \leq d_{\mathcal{E}}(q(h|\mathbf{x}), q(h|\mathbf{x}_i)) \leq K d(\mathbf{x}, \mathbf{x}_i).$$

Combined with Fubini's theorem, we obtain

$$\sum_{i=1}^{n} \mathbb{E}_{h \sim q(h|\mathbf{x}_i)} \mathbb{E}_{\mathbf{x} \sim \mu} \ell(h, \mathbf{x}) = \sum_{i=1}^{n} \mathbb{E}_{\mathbf{x} \sim \mu} \left[ \mathbb{E}_{h \sim q(h|\mathbf{x}_i)} \ell(h, \mathbf{x}) \right] \geq \sum_{i=1}^{n} \mathbb{E}_{\mathbf{x} \sim \mu} \left[ \mathbb{E}_{h \sim q(h|\mathbf{x})} \ell(h, \mathbf{x}) - K d(\mathbf{x}, \mathbf{x}_i) \right].$$

Combining this with Lemma B.1, yields Theorem 3.1. □

# C  Proofs of the results in Section 4

## C.1  Proof of Proposition 4.1

First, we recall the statement of Proposition 4.1.

**Proposition C.1** (Restatement of Proposition 4.1). *If there exists positive real numbers $K_\phi$ and $K_\theta$ such that the encoder and decoder are respectively $K_\phi$-Lipschitz and $K_\theta$-Lipschitz continuous, then*

$$d_{\mathcal{E}}\left(q_\phi(\mathbf{z}|\mathbf{x}_1), q_\phi(\mathbf{z}|\mathbf{x}_2)\right) \leq K_\phi K_\theta \left\| \mathbf{x}_1 - \mathbf{x}_2 \right\|, \tag{C.1}$$

*and*

$$\ell(\cdot, \mathbf{x}) \in \mathcal{E}, \quad \text{for any } \mathbf{x} \in \mathcal{X}. \tag{C.2}$$

*where $\mathcal{E} = \mathrm{Lip}_{K_\theta}(\mathcal{Z}, \mathbb{R})$ is the set of real-valued $K_\theta$-Lipschitz continuous functions defined on $\mathcal{Z}$.*

*Proof.*

1. Let us prove (C.1). First, since $q_\phi(\mathbf{z}|\mathbf{x}_i) = \mathcal{N}(\mu_\phi(\mathbf{x}_i), \mathrm{diag}(\sigma_\phi^2(\mathbf{x}_i)))$, by (A.2), the Wasserstein-2 distance $W_2(q_\phi(\mathbf{z}|\mathbf{x}_1), q_\phi(\mathbf{z}|\mathbf{x}_2))$ has the following closed form:

$$W_2(q_\phi(\mathbf{z}|\mathbf{x}_1), q_\phi(\mathbf{z}|\mathbf{x}_2))^2 = \left\| \mu_\phi(\mathbf{x}_1) - \mu_\phi(\mathbf{x}_2) \right\|^2 + \left\| \sigma_\phi(\mathbf{x}_1) - \sigma_\phi(\mathbf{x}_2) \right\|^2,$$

which, combined with the definition $Q_\phi(\mathbf{x}) = \begin{bmatrix} \mu_\phi(\mathbf{x}) \\ \sigma_\phi(\mathbf{x}) \end{bmatrix}$, yields

$$\left\| Q_\phi(\mathbf{x}_1) - Q_\phi(\mathbf{x}_2) \right\|^2 = W_2(q_\phi(\mathbf{z}|\mathbf{x}_1), q_\phi(\mathbf{z}|\mathbf{x}_2))^2.$$

Since $Q_\phi$ is $K_\phi$-Lipschitz continuous, we have $\left\| Q_\phi(\mathbf{x}_1) - Q_\phi(\mathbf{x}_2) \right\| \leq K_\phi \left\| \mathbf{x}_1 - \mathbf{x}_2 \right\|$, and

$$W_2(q_\phi(\mathbf{z}|\mathbf{x}_1), q_\phi(\mathbf{z}|\mathbf{x}_2)) \leq K_\phi \left\| \mathbf{x}_1 - \mathbf{x}_2 \right\|. \tag{C.3}$$

On the other hand, the definition $\mathcal{E} = \mathrm{Lip}_{K_\theta}(\mathcal{Z}, \mathbb{R})$ and the Kantorovich duality imply

$$d_{\mathcal{E}}(q_\phi(\mathbf{z}|\mathbf{x}_1), q_\phi(\mathbf{z}|\mathbf{x}_2)) = K_\theta W_1(q_\phi(\mathbf{z}|\mathbf{x}_1), q_\phi(\mathbf{z}|\mathbf{x}_2)).$$

Since $W_1 \leq W_2$, this equation, combined with (C.3) yields

$$d_{\mathcal{E}}(q_\phi(\mathbf{z}|\mathbf{x}_1), q_\phi(\mathbf{z}|\mathbf{x}_2)) \leq K_\theta K_\phi \left\| \mathbf{x}_1 - \mathbf{x}_2 \right\|.$$

2. Now, we shall prove (C.2), meaning, we show that $\ell(\cdot, \mathbf{x}) \in \mathrm{Lip}_{K_\theta}(\mathcal{Z}, \mathbb{R})$. Let $\mathbf{x} \in \mathcal{X}$ and $\mathbf{z}_1, \mathbf{z}_2 \in \mathcal{Z}$. We have

$$\begin{aligned}
\ell(\mathbf{z}_1, \mathbf{x}) - \ell(\mathbf{z}_2, \mathbf{x}) &= \left\| \mathbf{x} - g_\theta(\mathbf{z}_1) \right\| - \left\| \mathbf{x} - g_\theta(\mathbf{z}_2) \right\| \\
&= \left\| \mathbf{x} - g_\theta(\mathbf{z}_1) + g_\theta(\mathbf{z}_2) - g_\theta(\mathbf{z}_2) \right\| - \left\| \mathbf{x} - g_\theta(\mathbf{z}_2) \right\| \\
&\leq \left\| \mathbf{x} - g_\theta(\mathbf{z}_2) \right\| + \left\| g_\theta(\mathbf{z}_2) - g_\theta(\mathbf{z}_1) \right\| - \left\| \mathbf{x} - g_\theta(\mathbf{z}_2) \right\| \\
&= \left\| g_\theta(\mathbf{z}_2) - g_\theta(\mathbf{z}_1) \right\| \\
&\leq K_\theta \left\| \mathbf{z}_1 - \mathbf{z}_2 \right\|,
\end{aligned}$$

where the first inequality uses the triangle inequality and the second uses the Lipschitz assumption on $g_\theta$.

$\square$

## C.2 Proof of Theorem 4.3

*Proof of Theorem 4.3.* In order to prove Theorem 4.3, we need to upper bound the average distance and the exponential moment of Theorem 4.2, under the finite diameter assumption:

$$\sup_{\mathbf{x}, \mathbf{x}' \in \mathcal{X}} d(\mathbf{x}, \mathbf{x}') = \Delta < \infty. \tag{C.4}$$

More precisely, we need to prove the two following inequalities.

$$\sum_{i=1}^{n} \mathbb{E}_{\mathbf{x} \sim \mu} d(\mathbf{x}, \mathbf{x}_i) \leq n\Delta \tag{C.5}$$

and

$$n \log \mathbb{E}_{\mathbf{z} \sim p(\mathbf{z})} \mathbb{E}_{\mathbf{x} \sim \mu} \exp \left[ \frac{\lambda}{n} \left( \mathbb{E}_{\mathbf{x}' \sim \mu} \ell_{\mathrm{rec}}^{\theta}(\mathbf{z}, \mathbf{x}') - \ell_{\mathrm{rec}}^{\theta}(\mathbf{z}, \mathbf{x}) \right) \right] \leq \frac{\lambda^2 \Delta^2}{8n}. \tag{C.6}$$

First, (C.5) is a direct consequence of the definition of the diameter $\Delta$.

Now, let us prove (C.6). Let $\mathbf{z} \in \mathcal{Z}$. Since $\ell_{\mathrm{rec}}^{\theta}(\mathbf{z}, \mathbf{x}) = \|\mathbf{x} - g_{\theta}(\mathbf{z})\| = d(\mathbf{x}, g_{\theta}(\mathbf{z}))$ is the distance between $\mathbf{x}$ and $g_{\theta}(\mathbf{z})$, the definition of $\Delta$ implies $\ell_{\mathrm{rec}}^{\theta}(\mathbf{z}, \mathbf{x}) \in [0, \Delta]$, for any $\mathbf{x} \in \mathcal{X}$. Hence, applying Hoeffding's lemma on the random variables $\ell_i = \ell_{\mathrm{rec}}^{\theta}(\mathbf{z}, \mathbf{x}_i) \in [0, \Delta]$, we obtain

$$\mathbb{E}_{\mathbf{x} \sim \mu} \exp \left[ \frac{\lambda}{n} \left( \mathbb{E}_{\mathbf{x}' \sim \mu} \left[ \ell_{\mathrm{rec}}^{\theta}(\mathbf{z}, \mathbf{x}') \right] - \ell_{\mathrm{rec}}^{\theta}(\mathbf{z}, \mathbf{x}) \right) \right] \leq \exp \left[ \frac{\lambda^2 \Delta^2}{8n^2} \right].$$

Which leads to

$$n \log \mathbb{E}_{\mathbf{z} \sim p(\mathbf{z})} \mathbb{E}_{\mathbf{x} \sim \mu} \exp \left[ \frac{\lambda}{n} \left( \mathbb{E}_{\mathbf{x}' \sim \mu} \ell_{\mathrm{rec}}^{\theta}(\mathbf{z}, \mathbf{x}') - \ell_{\mathrm{rec}}^{\theta}(\mathbf{z}, \mathbf{x}) \right) \right] \leq n \log \mathbb{E}_{\mathbf{z} \sim p(\mathbf{z})} \exp \left[ \frac{\lambda^2 \Delta^2}{8n^2} \right] = \frac{\lambda^2 \Delta^2}{8n}.$$

$\square$

## C.3 Proof of Theorem 4.4

We need to bound the average distance and the exponential moment of Theorem 4.2, under the assumption $\mu = g^* \sharp p^*$, with $p^* = \mathcal{N}(\mathbf{0}, \mathbf{I})$ is the standard Gaussian distribution on $\mathbb{R}^{d^*}$, and $g^* \in \mathrm{Lip}_{K_*}(\mathbb{R}^{d^*}, \mathcal{X})$.

**Lemma C.2.** *Under the hypotheses of Theorem 4.4, the following inequality holds:*

$$n \log \mathbb{E}_{\mathbf{z} \sim p(\mathbf{z})} \mathbb{E}_{\mathbf{x} \sim \mu} \exp \left[ \frac{\lambda}{n} \left( \mathbb{E}_{\mathbf{x}' \sim \mu} \ell_{rec}^{\theta}(\mathbf{z}, \mathbf{x}') - \ell_{rec}^{\theta}(\mathbf{z}, \mathbf{x}) \right) \right] \leq \frac{\lambda^2 K_*^2}{2n}. \tag{C.7}$$

*Proof.* Let us show that

$$n \log \mathbb{E}_{\mathbf{z} \sim p(\mathbf{z})} \mathbb{E}_{\mathbf{x} \sim \mu} \exp \left[ \frac{\lambda}{n} \left( \mathbb{E}_{\mathbf{x}' \sim \mu} \ell_{\mathrm{rec}}^{\theta}(\mathbf{z}, \mathbf{x}') - \ell_{\mathrm{rec}}^{\theta}(\mathbf{z}, \mathbf{x}) \right) \right] \leq \frac{\lambda^2 K_*^2}{2n}.$$

Since $\mu = g^* \sharp p^*$, where $p^*$ is the standard Gaussian distribution on $\mathbb{R}^{d^*}$ and $g^*$ is $K_*$-Lipschitz continuous, the definition of the loss function $\ell_{\mathrm{rec}}^{\theta}$ implies

$$\mathbb{E}_{\mathbf{x} \sim \mu} \exp \left[ \frac{\lambda}{n} \left( \mathbb{E}_{\mathbf{x}' \sim \mu} \ell_{\mathrm{rec}}^{\theta}(\mathbf{z}, \mathbf{x}') - \ell_{\mathrm{rec}}^{\theta}(\mathbf{z}, \mathbf{x}) \right) \right]$$

$$= \mathbb{E}_{\mathbf{x} \sim \mu} \exp \left[ \frac{\lambda}{n} \left( \mathbb{E}_{\mathbf{x}' \sim \mu} \|\mathbf{x}' - g_{\theta}(\mathbf{z})\| - \|\mathbf{x} - g_{\theta}(\mathbf{z})\| \right) \right].$$

$$= \mathbb{E}_{\mathbf{w} \sim p^*} \exp \left[ \frac{\lambda}{n} \left( \mathbb{E}_{\mathbf{w}' \sim p^*} \|g^*(\mathbf{w}') - g_{\theta}(\mathbf{z})\| - \|g^*(\mathbf{w}) - g_{\theta}(\mathbf{z})\| \right) \right]$$

$$\overset{(*)}{\leq} \exp \left[ \frac{\lambda^2 K_*^2}{2n^2} \right].$$

This derivation implies

$$n \log \mathop{\mathbb{E}}_{\mathbf{z} \sim p(\mathbf{z})} \mathop{\mathbb{E}}_{\mathbf{x} \sim \mu} \exp \left[ \frac{\lambda}{n} \left( \mathop{\mathbb{E}}_{\mathbf{x}' \sim \mu} \ell_{\text{rec}}^\theta(\mathbf{z}, \mathbf{x}') - \ell_{\text{rec}}^\theta(\mathbf{z}, \mathbf{x}) \right) \right] \leq \frac{\lambda^2 K_*^2}{2n}.$$

We still need to justify $\overset{(*)}{\leq}$. Define for any arbitrary $\alpha \in \mathcal{X}$ the function $f : \mathbb{R}^{d^*} \to \mathbb{R}$ as:

$$f(\mathbf{w}) = \| g^*(\mathbf{w}) - \alpha \|.$$

Since $g^* \in \text{Lip}_{K_*}(\mathbb{R}^{d^*}, \mathcal{X})$, the function $f$ is $K_*$-Lipschitz. Indeed, for any $\mathbf{w}_1, \mathbf{w}_2 \in \mathbb{R}^{d^*}$,

$$\begin{aligned}
f(\mathbf{w}_1) - f(\mathbf{w}_2) &= \| g^*(\mathbf{w}_1) - \alpha \| - \| g^*(\mathbf{w}_2) - \alpha \| \\
&= \| g^*(\mathbf{w}_1) - \alpha + g^*(\mathbf{w}_2) - g^*(\mathbf{w}_2) \| - \| g^*(\mathbf{w}_2) - \alpha \| \\
&\leq \| g^*(\mathbf{w}_1) - g^*(\mathbf{w}_2) \| + \| g^*(\mathbf{w}_2) - \alpha \| - \| g^*(\mathbf{w}_2) - \alpha \| \\
&= \| g^*(\mathbf{w}_1) - g^*(\mathbf{w}_2) \| \\
&\leq K_* \| \mathbf{w}_1 - \mathbf{w}_2 \|
\end{aligned}$$

Moreover, it is known (see Theorem 5.5 of Boucheron et al. (2013)) that if $f$ is a $K_*$-Lipschitz function of a standard normal random variable $\mathbf{z}$, then

$$\mathbb{E} \, e^{\lambda(\mathbb{E}[f(\mathbf{z})] - f(\mathbf{z}))} \leq e^{\frac{\lambda^2 K_*^2}{2}}.$$

Hence,

$$\mathop{\mathbb{E}}_{\mathbf{w}_i \sim p^*} \left[ \exp \left[ \frac{\lambda}{n} \left( \mathop{\mathbb{E}}_{\mathbf{w}' \sim p^*} [\| g^*(\mathbf{w}') - g_\theta(\mathbf{z}) \|] - \| g^*(\mathbf{w}_i) - g_\theta(\mathbf{z}) \| \right) \right] \right] \leq \exp \left[ \frac{\lambda^2 K_*^2}{2n^2} \right],$$

which proves $\overset{(*)}{\leq}$ and concludes this proof. $\qquad\square$

**Lemma C.3.** *Under the hypotheses of Theorem 4.4, with probability at least $1 - \frac{nd^*}{2} e^{\frac{-a^2}{2}}$ over the random draw of S,*

$$\sum_{i=1}^{n} \mathop{\mathbb{E}}_{\mathbf{x} \sim \mu} d(\mathbf{x}, \mathbf{x}_i) \leq n K_* \sqrt{(1 + a^2) d^*} \tag{C.8}$$

*Proof.* First, since the training set $S = \{\mathbf{x}_1, \dots, \mathbf{x}_n\} \overset{\text{iid}}{\sim} \mu$, for each $1 \leq i \leq n$, there exists $\mathbf{w}_i \sim p^*$ such that $\mathbf{x}_i = g^*(\mathbf{w}_i)$. Let $a > 0$ be a positive real number. By definition of $p^*$, we have

$$\mathbb{P} \left[ \forall i, \mathbf{w}_i \in [-a, a]^{d^*} \right] = \left( \text{erf} \left( \frac{a}{\sqrt{2}} \right) \right)^{nd^*},$$

where $\text{erf}(\cdot)$ denotes the error function. Since the error function verifies (see Chu (1955))

$$\text{erf} \left( \frac{a}{\sqrt{2}} \right) \geq \sqrt{1 - e^{\frac{-a^2}{2}}},$$

we can use Bernoulli's inequality (see Section 2.4 of Mitrinovic and Vasic (1970)) to obtain

$$\mathbb{P} \left[ \forall i, \mathbf{w}_i \in [-a, a]^{d^*} \right] \geq \left( 1 - e^{\frac{-a^2}{2}} \right)^{nd^*/2} \geq 1 - \frac{nd^*}{2} e^{\frac{-a^2}{2}}. \tag{C.9}$$

Now we assume $\mathbf{w}_i \in [-a, a]^{d^*}$ for all $1 \leq i \leq n$ and we shall prove the desired inequality:

$$\sum_{i=1}^{n} \mathop{\mathbb{E}}_{\mathbf{x} \sim \mu} d(\mathbf{x}, \mathbf{x}_i) \leq n K_* \sqrt{(1 + a^2) d^*} \tag{C.10}$$

Let us prove (C.10). We have

$$\mathbb{E}_{\mathbf{x}\sim\mu} d(\mathbf{x},\mathbf{x}_i) = \mathbb{E}_{\mathbf{x}\sim\mu} \|\mathbf{x}-\mathbf{x}_i\| = \mathbb{E}_{\mathbf{w}\sim p^*} \|g^*(\mathbf{w})-g^*(\mathbf{w}_i)\| \le K_* \mathbb{E}_{\mathbf{w}\sim p^*} \|\mathbf{w}-\mathbf{w}_i\|, \qquad \text{(C.11)}$$

where the inequality follows from the assumption $g^* \in \mathrm{Lip}_{K_*}(\mathbb{R}^{d^*},\mathcal{X})$. Using Holder's inequality, the fact that $\|\mathbf{w}-\mathbf{w}_i\|^2$ is a non-central $\chi^2$ random variable with $d^*$ degrees of freedom and non-centrality coefficient $\|\mathbf{w}_i\|^2$, and the assumption $\mathbf{w}_i \in [-a,a]^{d^*}$, we obtain

$$\mathbb{E}_{\mathbf{w}\sim p^*} \|\mathbf{w}-\mathbf{w}_i\| \le \left( \mathbb{E}_{\mathbf{w}\sim p^*} \|\mathbf{w}-\mathbf{w}_i\|^2 \right)^{1/2} = \left( d^* + \|\mathbf{w}_i\|^2 \right)^{1/2} \le \left( d^* + a^2 d^* \right)^{1/2}.$$

Hence,

$$\mathbb{E}_{\mathbf{x}\sim\mu} \|\mathbf{x}-\mathbf{x}_i\| \le K_* \sqrt{(1+a^2)d^*}$$

which proves (C.10). $\qquad\square$

*Proof of Theorem 4.4.* Lemmas C.2 and C.3 applied to the result from Theorem 4.2 provide us with the inequality of Theorem 4.4. Finally, the confidence of $1-\delta - \frac{nd^*}{2}e^{\frac{-a^2}{2}}$ is obtained by using the union bound: the inequality in Theorem 4.2 holds with probability at least $1-\delta$, whereas the inequality appearing in Lemma C.3 holds with probability at least $1 - \frac{nd^*}{2}e^{\frac{-a^2}{2}}$. $\qquad\square$

In the following proposition, we provide an alternate version of Theorem 4.4, where the distribution $p^*$ is the uniform distribution[3] on $[0,1]^{d^*}$, instead of the standard Gaussian distribution on $\mathbb{R}^{d^*}$.

**Proposition C.4.** *Let $\mathcal{X}$ be the instance space, $\mathcal{Z}$ the latent space, $p(\mathbf{z}) \in \mathcal{M}_+^1(\mathcal{Z})$ the prior distribution, $\theta$ the parameters of the decoder, $\delta \in (0,1), \lambda > 0, a > 0$ be real numbers. Assume the data-generating distribution $\mu = g^* \sharp p^*$, where $p^* = \mathcal{U}([0,1]^{d^*})$ is the uniform distribution on $[0,1]^{d^*}$ and $g^* \in \mathrm{Lip}_{K_*}(\mathbb{R}^{d^*},\mathcal{X})$ is $K_*$-Lipschitz continuous. With probability at least $1-\delta$ over the random draw of $S$, the following holds for any posterior $q_\phi(\mathbf{z}|\mathbf{x})$:*

$$\mathbb{E}_{\mathbf{x}\sim\mu} \mathbb{E}_{q_\phi(\mathbf{z}|\mathbf{x})} \ell_{rec}^\theta(\mathbf{z},\mathbf{x}) - \frac{1}{n}\sum_{i=1}^n \left\{ \mathbb{E}_{q_\phi(\mathbf{z}|\mathbf{x}_i)} \ell_{rec}^\theta(\mathbf{z},\mathbf{x}_i) \right\} \le \frac{1}{\lambda} \left( \sum_{i=1}^n \mathrm{KL}(q_\phi(\mathbf{z}|\mathbf{x}_i) \,\|\, p(\mathbf{z})) + \right.$$
$$\left. \lambda K_\phi K_\theta K_* \sqrt{d^*} + \log\frac{1}{\delta} + \frac{\lambda^2 K_*^2}{2n} \right).$$

*Proof.* Let $\{\mathbf{w}_1,\dots,\mathbf{w}_n\} \subseteq [0,1]^{d^*}$ be such that for all $1 \le i \le n$, $\mathbf{x}_i = g^*(\mathbf{w}_i)$. Since the diameter of $[0,1]^{d^*}$ is $\sqrt{d^*}$, using the assumptions on $\mu$ and $g^*$, we obtain

$$\sum_{i=1}^n \mathbb{E}_{\mathbf{x}\sim\mu} d(\mathbf{x},\mathbf{x}_i) = \sum_{i=1}^n \mathbb{E}_{\mathbf{w}\sim p^*} d(g^*(\mathbf{w}),g^*(\mathbf{w}_i)) \le K_* \sum_{i=1}^n \mathbb{E}_{\mathbf{w}\sim p^*} \|\mathbf{w}-\mathbf{w}_i\| \le n K_* \sqrt{d^*}.$$

Applying the inequality above to Theorem 4.2 yields the desired result. $\qquad\square$

Note that unlike Theorem 4.4, the confidence $1-\delta$ of Theorem 4.2 is not lowered in Proposition C.4.

# D  Proofs of the results in Section 5

To simplify the proofs of the theorems of Section 5, we start by proving Lemmas D.1 and D.2 below. First, recall the definition of $\hat{\mu}_{\phi,\theta}$:

$$\hat{\mu}_{\phi,\theta} = \frac{1}{n}\sum_{i=1}^n g_\theta \sharp q_\phi(\mathbf{z}|\mathbf{x}_i).$$

The triangle inequality implies

$$W_1(\mu, g_\theta\sharp p(\mathbf{z})) \le W_1(\mu, \hat{\mu}_{\phi,\theta}) + W_1(\hat{\mu}_{\phi,\theta}, g_\theta\sharp p(\mathbf{z})). \qquad \text{(D.1)}$$

Let us state and prove the first lemma of this section.

---

[3]Note that the result holds for any distribution on $[0,1]^{d^*}$, not just the uniform distribution.

**Lemma D.1.** *The following inequality holds with probability at least $1 - \delta$ over the random draw of $S \sim \mu^{\otimes n}$:*

$$\lambda W_1(\mu, \hat{\mu}_{\phi,\theta}) \leq \frac{\lambda}{n} \sum_{i=1}^{n} \left( \underset{\mathbf{z} \sim q(\mathbf{z}|\mathbf{x}_i)}{\mathbb{E}} \ell_{rec}^{\theta}(\mathbf{z}, \mathbf{x}_i) \right) + \sum_{i=1}^{n} \text{KL}(q(\mathbf{z}|\mathbf{x}_i) \,||\, p(\mathbf{z})) +$$

$$\log \frac{1}{\delta} + \log \underset{S \sim \mu^{\otimes n}}{\mathbb{E}} \underset{\mathbf{z} \sim p(\mathbf{z})}{\mathbb{E}} e^{\lambda \left( \mathbb{E}_{\mathbf{x} \sim \mu} \left[ \ell_{rec}^{\theta}(\mathbf{z}, \mathbf{x}) \right] - \frac{1}{n} \sum_{i=1}^{n} \ell_{rec}^{\theta}(\mathbf{z}, \mathbf{x}_i) \right)}.$$

*Proof.* Recall the expression for the Wasserstein distance based on couplings:

$$W_1(\mu, \hat{\mu}_{\phi,\theta}) = \inf_{\pi \in \Gamma(\mu, \hat{\mu}_{\phi,\theta})} \int_{\mathcal{X} \times \mathcal{X}} \|\mathbf{x} - \mathbf{y}\| \, d\pi(\mathbf{x}, \mathbf{y})$$

In particular, $W_1(\mu, \hat{\mu}_{\phi,\theta})$ is less than the right-hand side obtained by the product coupling which can be rewritten, using Fubini's theorem, as:

$$W_1(\mu, \hat{\mu}_{\phi,\theta}) \leq \int_{\mathcal{X} \times \mathcal{X}} \|\mathbf{x} - \mathbf{y}\| \, d\mu(\mathbf{x}) d\hat{\mu}_{\phi,\theta}(\mathbf{y})$$

$$= \underset{\mathbf{y} \sim \hat{\mu}_{\phi,\theta}}{\mathbb{E}} \underset{\mathbf{x} \sim \mu}{\mathbb{E}} \|\mathbf{x} - \mathbf{y}\|.$$

Using the derivation above and the definition of $\hat{\mu}_{\phi,\theta}$, we obtain

$$W_1(\mu, \hat{\mu}_{\phi,\theta}) \leq \underset{\mathbf{y} \sim \hat{\mu}_{\phi,\theta}}{\mathbb{E}} \underset{\mathbf{x} \sim \mu}{\mathbb{E}} \|\mathbf{x} - \mathbf{y}\| = \frac{1}{n} \sum_{i=1}^{n} \left( \underset{\mathbf{z} \sim q_\phi(\mathbf{z}|\mathbf{x}_i)}{\mathbb{E}} \underset{\mathbf{x} \sim \mu}{\mathbb{E}} \|\mathbf{x} - g_\theta(\mathbf{z})\| \right)$$

$$= \frac{1}{n} \sum_{i=1}^{n} \left( \underset{\mathbf{z} \sim q_\phi(\mathbf{z}|\mathbf{x}_i)}{\mathbb{E}} \underset{\mathbf{x} \sim \mu}{\mathbb{E}} \ell_{\text{rec}}^{\theta}(\mathbf{z}, \mathbf{x}) \right).$$

We can upper bound this expression using Lemma B.1 with $\mathcal{H} = \mathcal{Z}$ and $\ell = \ell_{\text{rec}}^{\theta}$. We get that with probability at least $1 - \delta$ over the random draw of $S \sim \mu^{\otimes n}$:

$$\frac{\lambda}{n} \sum_{i=1}^{n} \left( \underset{\mathbf{z} \sim q(\mathbf{z}|\mathbf{x}_i)}{\mathbb{E}} \underset{\mathbf{x} \sim \mu}{\mathbb{E}} \ell_{\text{rec}}^{\theta}(\mathbf{z}, \mathbf{x}) \right) \leq \frac{\lambda}{n} \sum_{i=1}^{n} \left( \underset{\mathbf{z} \sim q(\mathbf{z}|\mathbf{x}_i)}{\mathbb{E}} \ell_{\text{rec}}^{\theta}(\mathbf{z}, \mathbf{x}_i) \right) + \sum_{i=1}^{n} \text{KL}(q(\mathbf{z}|\mathbf{x}_i) \,||\, p(\mathbf{z})) +$$

$$\log \frac{1}{\delta} + \log \underset{S \sim \mu^{\otimes n}}{\mathbb{E}} \underset{\mathbf{z} \sim p(\mathbf{z})}{\mathbb{E}} e^{\lambda \left( \mathbb{E}_{\mathbf{x} \sim \mu} \left[ \ell_{\text{rec}}^{\theta}(\mathbf{z}, \mathbf{x}) \right] - \frac{1}{n} \sum_{i=1}^{n} \ell_{\text{rec}}^{\theta}(\mathbf{z}, \mathbf{x}_i) \right)}.$$

$\square$

Therefore, using the upper bounds on the exponential moment from Section 4, we can prove Theorems 5.1 and 5.3 in the following sections.

Next, we prove the following lemma.

**Lemma D.2.** *The following inequality holds.*

$$W_1(\hat{\mu}_{\phi,\theta}, g_\theta \sharp p(\mathbf{z})) \leq \frac{K_\theta}{n} \sum_{i=1}^{n} \sqrt{\|\mu_\phi(\mathbf{x}_i)\|^2 + \left\|\sigma_\phi(\mathbf{x}_i) - \vec{1}\right\|^2},$$

*where $\vec{1} \in \mathbb{R}^{d_\mathcal{Z}}$ denotes the vector whose entries are all $1$.*

*Proof.* Defining the mixture of measures

$$\hat{q}_\phi(\mathbf{z}) = \frac{1}{n} \sum_{i=1}^{n} q_\phi(\mathbf{z}|\mathbf{x}_i),$$

the definition of $\hat{\mu}_{\phi,\theta}$ and the definition of a pushforward measures yield

$$\hat{\mu}_{\phi,\theta} = \frac{1}{n} \sum_{i=1}^{n} g_\theta \sharp q_\phi(\mathbf{z}|\mathbf{x}_i) = g_\theta \sharp \hat{q}_\phi(\mathbf{z}).$$

Using the dual formulation of the Wasserstein distance, we have

$$W_1(\hat{\mu}_{\phi,\theta}, g_\theta \sharp p(\mathbf{z})) = W_1\left(g_\theta \sharp \hat{q}_\phi(\mathbf{z}), g_\theta \sharp p(\mathbf{z})\right)$$

$$= \sup_{f \in \mathrm{Lip}_1(\mathcal{X},\mathbb{R})} \left[ \int_{\mathcal{Z}} f \circ g_\theta(\mathbf{z}) \, d\hat{q}_\phi(\mathbf{z}) - \int_{\mathcal{Z}} f \circ g_\theta(\mathbf{z}) \, dp(\mathbf{z}) \right]$$

$$= \sup_{g \in \mathcal{G}_\theta} \left[ \int_{\mathcal{Z}} g(\mathbf{z}) \, d\hat{q}_\phi(\mathbf{z}) - \int_{\mathcal{Z}} g(\mathbf{z}) \, dp(\mathbf{z}) \right]$$

$$\leq \sup_{g \in \mathrm{Lip}_{K_\theta}(\mathcal{Z},\mathbb{R})} \left[ \int_{\mathcal{Z}} g(\mathbf{z}) \, d\hat{q}_\phi(\mathbf{z}) - \int_{\mathcal{Z}} g(\mathbf{z}) \, dp(\mathbf{z}) \right]$$

$$= K_\theta W_1(\hat{q}_\phi(\mathbf{z}), p(\mathbf{z})),$$

where $\mathcal{G}_\theta = \{g : \mathcal{Z} \to \mathbb{R} \text{ s.t. } g = f \circ g_\theta \text{ and } f \in \mathrm{Lip}_1(\mathcal{X},\mathbb{R})\}$ and the inequality holds because $\mathcal{G}_\theta \subseteq \mathrm{Lip}_{K_\theta}(\mathcal{Z},\mathbb{R})$, since $g_\theta : \mathcal{Z} \to \mathcal{X}$ is $K_\theta$-Lipschitz. Now, since $(p,q) \mapsto W_1(p,q)$ is convex, the definition of $\hat{q}_\phi(\mathbf{z})$ implies

$$W_1(\hat{q}_\phi(\mathbf{z}), p(\mathbf{z})) \leq \frac{1}{n} \sum_{i=1}^{n} W_1(q_\phi(\mathbf{z}|\mathbf{x}_i), p(\mathbf{z})) \leq \frac{1}{n} \sum_{i=1}^{n} W_2(q_\phi(\mathbf{z}|\mathbf{x}_i), p(\mathbf{z})). \tag{D.2}$$

Since, by Equation (A.2),

$$W_2(q_\phi(\mathbf{z}|\mathbf{x}_i), p(\mathbf{z}))^2 = \|\mu_\phi(\mathbf{x}_i)\|^2 + \left\|\sigma_\phi(\mathbf{x}_i) - \vec{1}\right\|^2,$$

we obtain

$$W_1(\hat{\mu}_{\phi,\theta}, g_\theta \sharp p(\mathbf{z})) \leq \frac{K_\theta}{n} \sum_{i=1}^{n} \sqrt{\|\mu_\phi(\mathbf{x}_i)\|^2 + \left\|\sigma_\phi(\mathbf{x}_i) - \vec{1}\right\|^2}.$$

$\square$

## D.1 Proof of Theorem 5.1

*Proof of Theorem 5.1.* Recall from Lemma D.1 that with probability at least $1 - \delta$ over the random draw of $S \sim \mu^{\otimes n}$,

$$\lambda W_1(\mu, \hat{\mu}_{\phi,\theta}) \leq \frac{\lambda}{n} \sum_{i=1}^{n} \left( \mathop{\mathbb{E}}_{\mathbf{z} \sim q(\mathbf{z}|\mathbf{x}_i)} \ell_{\mathrm{rec}}^\theta(\mathbf{z}, \mathbf{x}_i) \right) + \sum_{i=1}^{n} \mathrm{KL}(q(\mathbf{z}|\mathbf{x}_i) \, \| \, p(\mathbf{z})) +$$
$$\log \frac{1}{\delta} + \log \mathop{\mathbb{E}}_{S \sim \mu^{\otimes n}} \mathop{\mathbb{E}}_{\mathbf{z} \sim p(\mathbf{z})} e^{\lambda\left( \mathbb{E}_{\mathbf{x} \sim \mu}[\ell_{\mathrm{rec}}^\theta(\mathbf{z},\mathbf{x})] - \frac{1}{n} \sum_{i=1}^{n} \ell_{\mathrm{rec}}^\theta(\mathbf{z},\mathbf{x}_i) \right)}. \tag{D.3}$$

In order to prove Theorem 4.3 in section C.2, we proved that

$$\mathop{\mathbb{E}}_{S \sim \mu^{\otimes n}} \exp\left[ \lambda \left( \mathop{\mathbb{E}}_{\mathbf{x} \sim \mu} \left[\ell_{\mathrm{rec}}^\theta(\mathbf{z},\mathbf{x})\right] - \frac{1}{n} \sum_{i=1}^{n} \ell_{\mathrm{rec}}^\theta(\mathbf{z},\mathbf{x}_i) \right) \right] \leq \exp\left[ \frac{\lambda^2 \Delta^2}{8n} \right].$$

Now, we can reuse this inequality to upper-bound the last term on the right-hand side of Equation (D.3). We obtain the desired theorem: under the assumptions of Theorem 4.3, with probability at least $1 - \delta$ over the random draw of $S \sim \mu^{\otimes n}$, the following holds for any posterior $q_\phi(\mathbf{z}|\mathbf{x})$:

$$W_1(\mu, \hat{\mu}_{\phi,\theta}) \leq \frac{1}{n} \sum_{i=1}^{n} \left\{ \mathop{\mathbb{E}}_{q_\phi(\mathbf{z}|\mathbf{x}_i)} \ell_{\mathrm{rec}}^\theta(\mathbf{z}, \mathbf{x}_i) \right\} + \frac{1}{\lambda} \left( \sum_{i=1}^{n} \mathrm{KL}(q_\phi(\mathbf{z}|\mathbf{x}_i) \, \| \, p(\mathbf{z})) + \log \frac{1}{\delta} + \frac{\lambda^2 \Delta^2}{8n} \right).$$

$\square$

## D.2 Proof of Theorem 5.2

*Proof of Theorem 5.2.* Theorem 5.2 is a direct consequence of Theorem 5.1 and Lemma D.2 applied to Equation (D.1). $\square$

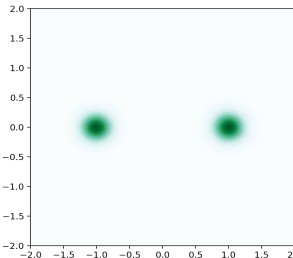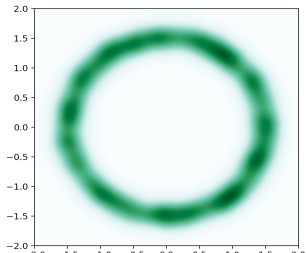

Figure 1: Samples from the real datasets

## D.3 Proof of Theorem 5.3

*Proof of Theorem 5.3.* Recall from Lemma D.1 that with probability at least $1 - \delta$ over the random draw of $S \sim \mu^{\otimes n}$,

$$\lambda W_1(\mu, \hat{\mu}_{\phi,\theta}) \leq \frac{\lambda}{n} \sum_{i=1}^{n} \left( \mathop{\mathbb{E}}_{\mathbf{z} \sim q(\mathbf{z}|\mathbf{x}_i)} \ell_{\text{rec}}^{\theta}(\mathbf{z}, \mathbf{x}_i) \right) + \sum_{i=1}^{n} \text{KL}(q(\mathbf{z}|\mathbf{x}_i) \| p(\mathbf{z})) +$$

$$\log \frac{1}{\delta} + \log \mathop{\mathbb{E}}_{S \sim \mu^{\otimes n}} \mathop{\mathbb{E}}_{\mathbf{z} \sim p(\mathbf{z})} e^{\lambda \left( \mathbb{E}_{\mathbf{x} \sim \mu} [\ell_{\text{rec}}^{\theta}(\mathbf{z}, \mathbf{x})] - \frac{1}{n} \sum_{i=1}^{n} \ell_{\text{rec}}^{\theta}(\mathbf{z}, \mathbf{x}_i) \right)}.$$

We can then use Lemma C.2 which stated that

$$\log \mathop{\mathbb{E}}_{\mathbf{z} \sim p(\mathbf{z})} \mathop{\mathbb{E}}_{S \sim \mu^{\otimes n}} \exp \left[ \lambda \left( \mathop{\mathbb{E}}_{\mathbf{x} \sim \mu} [\ell_{\text{rec}}^{\theta}(\mathbf{z}, \mathbf{x})] - \frac{1}{n} \sum_{i=1}^{n} \ell_{\text{rec}}^{\theta}(\mathbf{z}, \mathbf{x}_i) \right) \right] \leq \frac{\lambda^2 K_*^2}{2n}. \tag{D.4}$$

The expectations over $\mathbf{z}$ and $S$ can be swapped using Fubini's Theorem. Hence, combining Lemma C.2 and Lemma D.1, we obtain Theorem 5.3: with probability at least $1 - \delta$ over the random draw of $S \sim \mu^{\otimes n}$, the following holds for any posterior $q_{\phi}(\mathbf{z}|\mathbf{x})$.

$$W_1(\mu, \hat{\mu}_{\phi,\theta}) \leq \frac{1}{n} \sum_{i=1}^{n} \left\{ \mathop{\mathbb{E}}_{q_{\phi}(\mathbf{z}|\mathbf{x}_i)} \ell_{\text{rec}}^{\theta}(\mathbf{z}, \mathbf{x}_i) \right\} + \frac{1}{\lambda} \left( \sum_{i=1}^{n} \text{KL}(q_{\phi}(\mathbf{z}|\mathbf{x}_i) \| p(\mathbf{z})) + \log \frac{1}{\delta} + \frac{\lambda^2 K_*^2}{2n} \right).$$

$\square$

## D.4 Proof of Theorem 5.4

*Proof of Theorem 5.4.* Theorem 5.4 is a direct consequence of Theorem 5.3 and Lemma D.2 applied to Equation (D.1). $\square$

## E  Numerical Experiments

We computed the numerical value of the bound of Theorem 4.3. We performed the experiments on two 2-dimensional synthetic datasets. The first one is a mixture of two isotropic Gaussian distributions on $\mathbb{R}^2$ centered at $(-1, 0)$ and $(1, 0)$ respectively, and with standard deviation $\sigma = 0.1$ and null covariances. The second dataset consists of noisy samples arranged in a circle centered at the origin, with radius 1.5 and standard deviation $\sigma = 0.1$. Both datasets are truncated so that no sample is over 4 standard deviations away from its corresponding mean. This is to formally ensure that the diameter of the instance spaces is finite, as required by Theorem 4.3. The sizes of the training, validation and test sets are respectively 50,000, 20,000 and 20,000. Samples from the two datasets are shown in Figure 1.

We used the same architecture and hyperparameters for both datasets. The encoder and decoder are fully connected networks with 3 hidden layers and 100 hidden units per layer. We also set the Lipschitz constants of the encoder and decoder networks to $K_{\phi} = K_{\theta} = 2$. In order to enforce

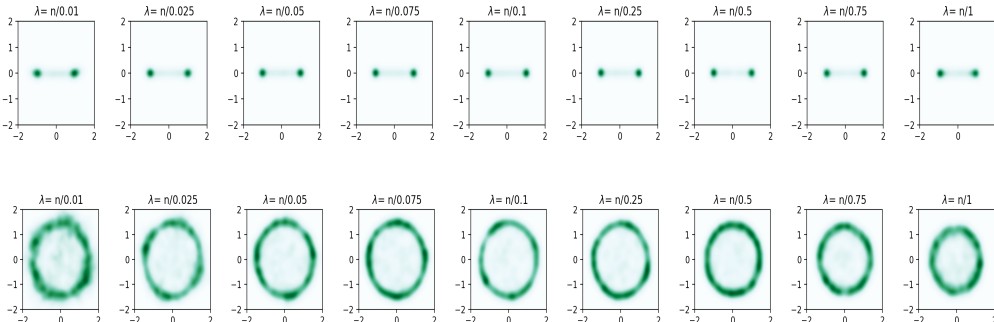

Figure 2: Samples from the models trained on the 2-Gaussian dataset (top) and the Circle dataset (bottom).

| $\lambda$ | **Test Rec. loss** | Emp. Rec. loss | Emp. KL loss | Exp. moment | **Bound** |
|---|---|---|---|---|---|
| $n/0.01$ | 0.1107 | 0.1110 | 0.0192 | 89.00 | 99.80 |
| $n/0.025$ | 0.1228 | 0.1237 | 0.0505 | 35.60 | 46.45 |
| $n/0.05$ | 0.1299 | 0.1299 | 0.1010 | 17.80 | 28.70 |
| $n/0.075$ | 0.1388 | 0.1403 | 0.1511 | 11.867 | 22.83 |
| $n/0.1$ | 0.1425 | 0.1436 | 0.2003 | 8.900 | 19.92 |
| $n/0.25$ | 0.1707 | 0.1732 | 0.4883 | 3.560 | 14.89 |
| $n/0.5$ | 0.2120 | 0.2162 | 0.9602 | 1.780 | 13.63 |
| $n/0.75$ | 0.2718 | 0.2725 | 1.4122 | 1.1868 | 13.54 |
| $n/1$ | 0.3586 | 0.3596 | 1.8593 | 0.8901 | 13.78 |

Table 1: Table showing the values of the different quantities of Equation E.1 for the "2-Gaussian" dataset. The upper bound on the average distance term is 10.67.

Lipschitz continuity, we used Björck orthonormalization (Björck and Bowie, 1971) with GroupSort activations (Anil et al., 2019), and we utilized the implementation of Lipschitz layers by Anil et al. (2019). Note that Barrett et al. (2022) performed experiments with VAEs with fixed Lipschitz constants, but we did not directly use their implementation because of a difference in the definition of the Lipschitz norm of the encoder, which affects the implementation. Note also that unlike the usual computations of PAC-Bayesian bounds (Pérez-Ortiz et al., 2021), our implementation does not use probabilistic neural networks. It uses deterministic networks, as it is usual for VAEs, because our analysis did not include additional stochasticity. We used the MSE as the reconstruction loss during training, and computed the bounds on validation datasets. The samples from the different models are displayed in Figure 2.

Recall the inequality of Theorem 4.3:

$$
\underbrace{\mathbb{E}_{\mathbf{x}\sim\mu}\mathbb{E}_{q_\phi(\mathbf{z}|\mathbf{x})}\ell_{\text{rec}}^\theta(\mathbf{z},\mathbf{x})}_{\text{Test Rec. Loss}} \leq \underbrace{\frac{1}{n}\sum_{i=1}^{n}\left\{\mathbb{E}_{q_\phi(\mathbf{z}|\mathbf{x}_i)}\ell_{\text{rec}}^\theta(\mathbf{z},\mathbf{x}_i)\right\}}_{\text{Emp. Rec. Loss}} + \underbrace{\frac{1}{\lambda}\sum_{i=1}^{n}\text{KL}(q_\phi(\mathbf{z}|\mathbf{x}_i)\,\|\,p(\mathbf{z}))}_{\text{Emp. KL loss}} +
$$
$$
\underbrace{K_\phi K_\theta \Delta}_{\text{Avg distance}} + \underbrace{\frac{\lambda\Delta^2}{8n}}_{\text{Exp. moment}} + \frac{1}{\lambda}\log\frac{1}{\delta}. \tag{E.1}
$$

Tables 1 and 2 show the numerical values of the bound of Theorem 4.3 for different values of $\lambda$. The first column is approximated using the test set, and the last one refers to all the right-hand side of (E.1). The empirical reconstruction and KL losses are computed using the validation set, since, as mentioned in the main paper, the bounds need to be computed using a set independent from the training set.

| $\lambda$ | **Test Rec. loss** | Emp. Rec. loss | Emp. KL loss | Exp. moment | **Bound** |
|---|---|---|---|---|---|
| $n/0.01$ | 0.095 | 0.0959 | 0.0197 | 180.50 | 195.81 |
| $n/0.025$ | 0.1354 | 0.1362 | 0.0525 | 72.20 | 87.59 |
| $n/0.05$ | 0.1785 | 0.1783 | 0.1058 | 36.10 | 51.58 |
| $n/0.075$ | 0.2005 | 0.2020 | 0.1587 | 24.07 | 39.63 |
| $n/0.1$ | 0.2245 | 0.2247 | 0.2117 | 18.05 | 33.69 |
| $n/0.25$ | 0.3498 | 0.3486 | 0.5160 | 7.220 | 23.28 |
| $n/0.5$ | 0.5026 | 0.4940 | 0.9997 | 3.610 | 20.30 |
| $n/0.75$ | 0.6171 | 0.6154 | 1.4691 | 2.406 | 19.691 |
| $n/1$ | 0.7513 | 0.7499 | 1.9314 | 1.805 | 19.686 |

Table 2: Table showing the values of the different quantities of Equation E.1 for the "Circle" dataset. The upper bound on the average distance term is 15.2.

From Tables 1 and 2, once can see that the bounds are dominated by two terms: the average distance and the exponential moment. Although as $\lambda$ approaches $n$, the exponential moment gets smaller and the main influence comes from the upper bound on the average distance. Hence, in order to tighten the bound, one may need to derive tighter upper bounds on the average distance, or derive versions of Theorem 4.3 where this term is replaced by a numerically smaller one.

# F Additional Results and Remarks

This section contains additional remarks and discussions. We start with possible extensions of our results.

## F.1 The variance of the likelihood

Our definition of the decoder network's output (the function $g_\theta : \mathcal{Z} \to \mathcal{X}$) only considers the deterministic part of the decoder. In other words, our results only apply to VAEs whose likelihood has constant variance. However, they can be extended to cases when the variance of the likelihood is optimized, but at a cost. We discuss separately the two cases where the variance depends on individual datapoints or not.

**Instance-independent variance.** If the standard deviation $\sigma$ of the decoder is fixed, then we have $\sigma \propto \frac{n}{\lambda}$, (recall the hyperparameter $\lambda$ from Theorem 3.1 and subsequent theorems). Hence, optimizing $\sigma$ corresponds to optimizing $\lambda$, which is non-trivial in PAC-Bayes. Indeed, most PAC-Bayes bounds (including ours) do not directly allow one to optimize $\lambda$ (see Section 2.1.4 of Alquier (2021)). Although there are some ways around this restriction, we are not aware of any results that allow one to optimize in the general case (meaning continuous values of $\lambda$ and unbounded loss). For $[0, 1]$-bounded loss functions, Thiemann et al. (2017) developed a PAC-Bayes bound uniformly valid for a trade-off parameter $\lambda'$, and show that one can optimize w.r.t. both the posterior and $\lambda'$, under certain assumptions. For unbounded losses, if one assumes $\lambda \in \Lambda$, where $|\Lambda|$ is finite, a union bound argument allows one to make the bound uniform with respect to $\lambda$, at the cost of $\log |\Lambda|$ (see Alquier (2021)). One can still optimize with respect to a continuous set $\Lambda$, by considering a grid. For instance, if one considers $\Lambda \cap \{1, \dots, n\}$, then the penalty is $\log n$ and if one considers $\Lambda \cap \{e^k : 1 \le k \le n\}$, the penalty is $\log \log n$.

**Instance-dependent variance.** Now, assume the standard deviation is dependent on individual instances. Say we define the reconstruction loss as $\ell_\theta(\mathbf{z}, \mathbf{x}) = \frac{1}{\sigma_\theta(\mathbf{z})} \|\mathbf{x} - g_\theta(\mathbf{z})\|$, where $\sigma_\theta : \mathcal{Z} \to \mathbb{R}_{>0}$. Because of the division by $\sigma_\theta(\mathbf{z})$, let us assume that there is a fixed upper bound $\sigma_1 > 0$ such that $\sigma_\theta(\mathbf{z}) > \sigma_1$, for any $\mathbf{z} \in \mathcal{Z}$. There are two main tasks: making sure Assumption 1 is satisfied, and bounding the exponential moment of Theorem 4.2, with this new loss function.

Verifying Assumption 1 is equivalent to showing that Proposition 4.1 is verified for this new loss function $\ell_\theta$. The second part of the proof of Proposition 4.1 tells us that we need to show that $\ell_\theta$ is Lipschitz-continuous. Note that in general, the product of real-valued Lipschitz functions is not Lipschitz. Hence, we assume, in addition, that $\|\mathbf{x} - g_\theta(\mathbf{z})\| \le M < \infty$. The following proposition shows that Assumption 1 is satisfied with the constant $K = K_\phi \left( \frac{K_\sigma M}{\sigma_1^2} + \frac{K_\theta}{\sigma_1} \right)$.

**Proposition F.1.** *Consider a VAE with parameters $\phi$ and $\theta$ and let $K_\phi, K_\theta \in \mathbb{R}$ be the Lipschitz norms of the encoder and decoder respectively. Also, consider the loss function $l_{rec}^\theta : \mathcal{Z} \times \mathcal{X} \to \mathbb{R}$ defined as*

$$l_{rec}^\theta(\mathbf{z}, \mathbf{x}) = \frac{1}{\sigma_\theta(\mathbf{z})} \|\mathbf{x} - g_\theta(\mathbf{z})\|$$

*where $\sigma_\theta : \mathcal{Z} \to \mathbb{R}_{>0}$ is $K_\sigma$-Lipschitz. Assume and for all $\mathbf{z} \in \mathcal{Z}$, $\sigma_\theta(\mathbf{z}) > \sigma_1$ and $\|\mathbf{x} - g_\theta(\mathbf{z})\| \le M$ for some fixed $0 < \sigma_1 < 1$ and $M > 0$. Then the variational distribution $q_\phi(\mathbf{z}|\mathbf{x})$ satisfies Assumption 1 with $\mathcal{E} = \{ f : \mathcal{Z} \to \mathbb{R} : \|f\|_{Lip} \le \frac{K_\sigma M}{\sigma_1^2} + \frac{K_\theta}{\sigma_1} \}$, $K = K_\phi \left( \frac{K_\sigma M}{\sigma_1^2} + \frac{K_\theta}{\sigma_1} \right)$, and $\ell = l_{rec}^\theta$.*

*Proof.* The first part of Assumption 1 is satisfied, since $\frac{K_\sigma M}{\sigma_1^2} + \frac{K_\theta}{\sigma_1} > K_\theta$. Now, for the second part of Assumption 1, we need to show that $l_{rec}^\theta$ is $\frac{K_\sigma M}{\sigma_1^2} + \frac{K_\theta}{\sigma_1}$-Lipschitz continuous. First,

$$\left| \frac{1}{\sigma_\theta(\mathbf{z}_1)} - \frac{1}{\sigma_\theta(\mathbf{z}_2)} \right| = \left| \frac{\sigma_\theta(\mathbf{z}_2) - \sigma_\theta(\mathbf{z}_1)}{\sigma_\theta(\mathbf{z}_1)\sigma_\theta(\mathbf{z}_2)} \right| \le \frac{K_\sigma \|\mathbf{z}_1 - \mathbf{z}_2\|}{\sigma_1^2}.$$

We have

$$\left| l_{rec}^\theta(\mathbf{z}_1, \mathbf{x}) - l_{rec}^\theta(\mathbf{z}_2, \mathbf{x}) \right| = \left| \frac{1}{\sigma_\theta(\mathbf{z}_1)} \|\mathbf{x} - g_\theta(\mathbf{z}_1)\| - \frac{1}{\sigma_\theta(\mathbf{z}_2)} \|\mathbf{x} - g_\theta(\mathbf{z}_2)\| \right|$$

$$= \left| \frac{1}{\sigma_\theta(\mathbf{z}_1)} - \frac{1}{\sigma_\theta(\mathbf{z}_2)} \right| \|\mathbf{x} - g_\theta(\mathbf{z}_1)\| + \frac{1}{\sigma_\theta(\mathbf{z}_2)} \left| \|\mathbf{x} - g_\theta(\mathbf{z}_1)\| - \|\mathbf{x} - g_\theta(\mathbf{z}_2)\| \right|$$

$$\le \frac{K_\sigma M}{\sigma_1^2} \|\mathbf{z}_1 - \mathbf{z}_2\| + \frac{K_\theta}{\sigma_1} \|\mathbf{z}_1 - \mathbf{z}_2\|$$

$$= \left( \frac{K_\sigma M}{\sigma_1^2} + \frac{K_\theta}{\sigma_1} \right) \|\mathbf{z}_1 - \mathbf{z}_2\|$$

$\square$

Now, let us focus on bounding the exponential moment. In this case, when the instance space is bounded, the upper bound on the exponential moment (in the proof of Theorem 4.3) is:

$$\frac{\lambda^2 \Delta^2}{8n\sigma_1^2}, \quad \text{instead of} \quad \frac{\lambda^2 \Delta^2}{8n}.$$

And under the manifold assumption, we get the following upper bound (in the proof of Theorem 4.4):

$$\frac{\lambda^2 K_*^2}{2n\sigma_1^2}, \quad \text{instead of} \quad \frac{\lambda^2 K_*^2}{2n}$$

Note that although the upper bounds on the average distance remain unchanged, the coefficient $K_\phi K_\theta$ is replaced by $K_\phi \left( \frac{K_\sigma M}{\sigma_1^2} + \frac{K_\theta}{\sigma_1} \right)$, which is larger, specially if $\sigma_1$ is very small.

### F.2 Uniformity with respect to $\theta$

As mentioned in the main paper, although our bounds hold uniformly for any encoder $\phi$, they only hold for a given decoder $\theta$. the consequence of this limitation is that the numerical computations of the bounds need to be done on a sample set disjoint from the training set (e.g. a validation or test set). Let $\Theta$ denote a set of decoder parameters over which the optimization is performed.

From a theoretical perspective, the union bound can be used to circumvent this issue, when we consider a finite set of parameters $\Theta$. In that case, the $\log \frac{1}{\delta}$ in Theorem 3.1 becomes $\log \frac{|\Theta|}{\delta}$, which

loosens the bound. Moreover, since $\Theta$ denotes a set of neural network parameters, this assumption may not be appropriate unless one chooses a very large set $\Theta$, which can significantly loosen the bound.

Another option would be to make assumptions on the complexity of the set of loss functions $\{\ell_{\text{rec}}^\theta : \theta \in \Theta\}$ parameterized by decoder parameters $\theta \in \Theta$ (e.g. the Rademacher complexity), in order to obtain uniform bounds in a more general case. We leave such explorations to future works.

### F.3  Additional Remarks

**Remark F.1** (Alternate formulation of Assumption 1). We can provide an equivalent formulation of Assumption 1. A posterior $q(h|\mathbf{x})$ and a loss function $\ell$ satisfy Assumption 1 with a constant $K > 0$ if and only if for any $\mathbf{x} \in \mathcal{X}$,

$$\left| \mathop{\mathbb{E}}_{h \sim q(h|\mathbf{x}_1)} \ell(h, \mathbf{x}) - \mathop{\mathbb{E}}_{h \sim q(h|\mathbf{x}_2)} \ell(h, \mathbf{x}) \right| \le K d(\mathbf{x}_1, \mathbf{x}_2).$$

The formulation given in the paper is more intuitive, but this expression shows that the specific choice of $\mathcal{E}$ does not matter. The equivalence of the two formulations is a consequence of the definition of an IPM.

**Remark F.2** (Prior Learning in PAC-Bayes). The majority of PAC-Bayesian bounds (McAllester, 1999; Seeger, 2002; Germain et al., 2009; Mbacke et al., 2023) require the prior distribution $p$ on the hypothesis class to be independent of the training set[4]. In practice, this means one has to use data-free priors when minimizing PAC-Bayes bounds. Since, in that case, the learned posterior is likely very far from the prior, the KL-divergence tends to be orders of magnitude larger than the empirical risk. In practice, this means the optimization is monopolized by the KL-divergence, leading to a poor performance of the learning algorithm. In order to avoid this issue and still obtain a valid certificate, the following "prior learning trick" is used. Split the training set $S = \{\mathbf{x}_1, \ldots, \mathbf{x}_n\}$ in two disjoint subsets $S_1, S_2$, where $|S_1| = n_0, |S_2| = n - n_0$ with $n_0 < n$. Then, learn the prior $p$ on $S_1$, learn the posterior $q$ on $S$ (the whole training set), and compute the certificate on $S_2$.

The reason why this trick cannot be directly applied to circumvent the fact that our bounds are valid for a given decoder, is that the encoder and the decoder are jointly optimized in VAEs. Hence, one has to make sure the samples used to learn the encoder (hence, train the model) are not used in the computation of the risk certificate. We emphasize that in our case, the issue does not lie in the learning of the prior (the standard VAE considers a standard Gaussian prior), but of the loss function $\ell_{\text{rec}}^\theta$, which is dependent on the decoder's parameters $\theta$.

---

[4]PAC-Bayesian bounds with data-dependent priors were developed by Dziugaite and Roy (2018); Rivasplata et al. (2020).