# OpenReview forum: "Statistical Guarantees for Variational Autoencoders using PAC-Bayesian Theory"
_NeurIPS.cc/2023/Conference — NeurIPS 2023 spotlight_

### Official Review · Reviewer_Hutj · 2023-06-20

**Soundness:** 3 good
**Presentation:** 4 excellent
**Contribution:** 3 good
**Rating:** 7
**Confidence:** 2

**Summary:**

The paper introduces generalization bounds for variational autoencoders (VAEs) by employing PAC-Bayesian bounds. Initially, the authors derive a PAC-Bayesian bound utilizing a posterior distribution (Theorem 3.1). This result is subsequently utilized to establish a generalization bound for the reconstruction loss (Theorem 4.2) and the generated distribution of the VAE (Theorem 5.1). These two outcomes rely on relatively broad quantities, which are further elaborated in corollaries that incorporate additional assumptions, namely bounded instance spaces and manifold assumptions.


**Strengths:**

The paper is well written, presenting a notable and original scientific contribution to the field. The assumptions are thoroughly discussed, and despite the technical nature of the results, the authors make an effort to provide the reader with intuitive explanations, which is commendable.


**Weaknesses:**

Considering the nature of the NeurIPS conference, it would be beneficial to include an experimental section in the paper. Given its strong theoretical content, conducting experiments on synthetic problems could be valuable in assessing the asymptotic behavior of the bound concerning various parameters such as reconstruction losses, $\lambda$, and Lipschitz constants. For example, these synthetic experiments could utilize the assumption of bounded instance space, with a 1D input space, allowing for accurate approximation of the Lipschitz constants of the models.

Aside from this suggestion, as I am not familiar with PAC-Bayes theory or the VAE literature, I am unable to identify any evident flaws or weaknesses in the paper that could aid the authors in improving their work.

**Questions:**

* Does the dimension of the latent space has any direct influence on the bound, or is it present only through e.g. the space's diameter?

**Limitations:**

The authors mention the limitation of their approach in the conclusion.

---

> ### Author Rebuttal · Authors · 2023-08-06
>
> We thank the reviewer for their thoughtful review and positive assessment of our work.
>
> > Given its strong theoretical content, conducting experiments on synthetic problems could be valuable in assessing the asymptotic behavior of the bound
>
> Given the technical nature of the results, our focus was on making the presentation as clear and intuitive as possible, without sacrificing mathematical accuracy. We address the question of numerical experiments in our general response.
>
> > Does the dimension of the latent space has any direct influence on the bound, or is it present only through e.g. the space's diameter?
>
> This is a great question. Although it does not explicitly appear in the expressions, the dimension $d\_\\mathcal{Z}$ of the latent space does have an influence on the bounds, because it affects both the reconstruction loss, and the discrepancy between $q\_\\phi(\\mathbf{z} | \\mathbf{x}\_i)$ and $p(\\mathbf{z})$. Indeed, if $d\_\\mathcal{Z}$ is too small, then depending on the complexity of the encoder and decoder networks, the model may not be able to properly reconstruct the samples, which leads to larger empirical and population reconstruction losses. On the other hand, if $d\_\\mathcal{Z}$ is too large, then the KL divergence may get too large as well, and if $q\_\\phi(\\mathbf{z} | \\mathbf{x}\_i)$ and $p(\\mathbf{z})$ are too far apart, then the upper bounds on the Wasserstein distance between $\\mu$ and $g\_\\theta \sharp p(\\mathbf{z})$ becomes larger, because they depend on the Wasserstein-2 distance between $q\_\\phi(\\mathbf{z} | \\mathbf{x}_i)$ and $p(\\mathbf{z})$.
>
> Note that in our bounds leveraging the manifold assumption (Theorems 4.4, 5.3 and 5.4), the intrinsic dimension $d^*$ can be different from the latent dimension $d\_\\mathcal{Z}$. The intrinsic dimension explicitly appears in the bound of Theorem 4.4, but not Theorems 5.3 and 5.4. This is because the upper-bound on the exponential moment is dimension-free, and the bounds of Section 5 do not depend on the average distance.
>
> We thank the reviewer again for their hard work and insightful review.

---

> > ### Comment · Reviewer_Hutj · 2023-08-18
> >
> > Thanks to the author for answering my question. I maintain my score.

---

### Official Review · Reviewer_3pjC · 2023-07-01

**Soundness:** 3 good
**Presentation:** 4 excellent
**Contribution:** 3 good
**Rating:** 7
**Confidence:** 3

**Summary:**

This paper derives novel PAC-Bayesian bounds for VAEs by treating the variational posterior as the PAC-Bayes posterior. To do this, the authors must adapt the PAC-Bayes theorem to also hold in the case where the posterior is conditioned on a learning sample.

**Strengths:**

- Well-written paper, with well presented results
- Limitations clearly stated without exaggeration
- Clearly addresses an important theoretical topic
- Novel theoretical results

**Weaknesses:**

- Bound unfortunately needs to be computed using samples different from those used to train the VAE. This runs counter to one of the most interesting and useful aspects of PAC-Bayes bounds - that they can be used as learning objectives.
- The bounds are not numerically computed.

**Questions:**

- Have the authors tried numerically computing these bounds in any situations? How tight are they? How far are they from being non-vacuous?

**Limitations:**

The authors describe the limitations well in their Discussion and Conclusion section, which is well-written and insightful.

---

> ### Author Rebuttal · Authors · 2023-08-06
>
> We thank the reviewer for their insightful comments and their appreciation of our theoretical contributions.
>
> > Bound unfortunately needs to be computed using samples different from those used to train the VAE. This runs counter to one of the most interesting and useful aspects of PAC-Bayes bounds - that they can be used as learning objectives.
>
> The reviewer is right. As mentioned in the conclusion, the bounds need to be computed using a set of samples disjoint from the training set. We note that since the results in Section 5 provide upper bounds on the Wasserstein distance between distributions on (possibly) high-dimensional spaces, having empirical upper bounds may still be very useful, because of the difficulty of estimating the Wasserstein distance in high dimension in the general case.
>
> > Have the authors tried numerically computing these bounds in any situations? How tight are they? How far are they from being non-vacuous?
>
> We are currently working on experiments on synthetic datasets, and we will add the results to the manuscript. We also address the question of the experiments in our general response.
>
> Once again, we thank the reviewer for their hard work and insightful review.

---

> > ### Comment · Reviewer_3pjC · 2023-08-17
> >
> > Thanks for your response. As it seems my understanding of the situation is correct, I will maintain my score.

---

### Official Review · Reviewer_8cep · 2023-07-02

**Soundness:** 3 good
**Presentation:** 3 good
**Contribution:** 3 good
**Rating:** 7
**Confidence:** 4

**Summary:**

In this paper, the authors provide a novel general PAC-Bayesian bound for a posterior distribution conditioned on individual elements of the instance space (and not only on observed samples). They then use it to derive generalization bounds for reconstruction loss, regeneration, and generation in VAE (both for bounded instance space and under the manifold assumption).

**Strengths:**

1. The results of the paper appear to be original and significant. As the authors claim, this is probably the first work that provides statistical guarantees for VAE.

2. The paper is clearly written, so it is easy to understand.

3. The paper contains a formal theoretical analysis and an intuitive discussion of assumptions and results.

**Weaknesses:**

1. Perhaps the main weakness is the lack of experimental results. However, I am not sure they are necessary (compare, e.g., with the related work [1]).

[1] Chakrabarty, A. and Das, S. (2021). Statistical regeneration guarantees of the Wasserstein autoencoder with latent space consistency. In Advances in Neural Information Processing Systems.

**Questions:**

Minor comments:

l. 46: I would be careful about calling WAE a variant of VAE.

l. 92: $p\ll q$ could be explained.

l. 106: Is "data generating distribution" the same as "input distribution"?

l. 134: footnote character '1' should be before a comma.

l. 170 (bottom): ',' --> '.'.

l. 192: '.' --> ':'.

l. 219: "exists" --> "exist".

l. 289: Is the word "uniform" necessary here?

l. 321, 327: Add ':' at the end of the line.

**Limitations:**

The authors adequately addressed the limitations of the paper.

---

> ### Author Rebuttal · Authors · 2023-08-06
>
> We thank the reviewer for their positive feedback and thoughtful suggestions, which will help us improve the paper.
>
> > Perhaps the main weakness is the lack of experimental results. However, I am not sure they are necessary (compare, e.g., with the related work [1])
>
> Indeed, the main objective of this work is theoretical. [1] present asymptotic bounds on the regeneration properties of WAEs, while our bounds are empirical, and cover the reconstruction, regeneration, and generation properties of VAEs. We also address the question of the experiments in our general response.
>
> > I would be careful about calling WAE a variant of VAE.
>
> We understand how this could be bothersome. We will rewrite that sentence accordingly.
>
> > l. 106: Is "data generating distribution" the same as "input distribution"?
>
> Yes, they are the same thing. Both designations refer to the distribution later denoted $\mu$.
>
> > l. 289: Is the word "uniform" necessary here?
>
> Yes, it is necessary, and we agree that the sentence is a bit ambiguous. Here, the word "uniform" refers to the choice of $ i \\in \\{1, \\dots, n \\}$, such that the distribution $q\_\\phi(\\mathbf{z} | \\mathbf{x}\_{i} )$ is used to sample $\\mathbf{z} \\sim q\_\\phi(\\mathbf{z} | \\mathbf{x}\_i)$.
> In other words, it is not $\\mathbf{z}$ that is sampled "uniformly", but $ i \\in \\{ 1, \\dots, n \\} $ that is sampled uniformly, since all the coefficients in the empirical regenerated distribution are equal to $\\frac{1}{n}$. We will reformulate the sentence to eliminate the ambiguity.
>
> Once again, we thank the reviewer for their hard work. We are grateful for the reviewer's suggestions, and will use them to improve the manuscript.

---

> > ### Comment · Reviewer_8cep · 2023-08-15
> > **Thank you for the response**
> >
> > I am satisfied with the authors' rebuttal. I am willing to raise my rating depending on the conclusions of the second phase of the discussion.

---

### Official Review · Reviewer_sohu · 2023-07-06

**Soundness:** 3 good
**Presentation:** 3 good
**Contribution:** 3 good
**Rating:** 6
**Confidence:** 3

**Summary:**

【Post-rebuttal Comments】
I thank the authors for the discussions after the authors' rebuttal. My questions about variance estimation are appropriately answered. So, I want to keep my score and vote for acceptance.

【Original Comments】
This paper derives the PAC-Bayes bound on the hypothesis set by conditional distributions (Theorem 3.1). As an example of its application, this paper gives three kinds of generalization bounds for Variational AutoEncoder by interpreting its encoder as a hypothesis set. Specifically, this paper provides guarantees for the reconstruction of data points (Theorem 4.3), the regeneration of data distributions (Theorem 5.1), and the generation from prior distributions (Theorem 5.2). Furthermore, by assuming the manifold hypothesis for the input distribution, sharper bounds are derived that depend on the manifold's dimension rather than the input dimension (Theorem 4.4, Theorem 5.3, Theorem 5.4).

**Strengths:**

- Instead of simply applying the existing PAC-Bayes bound, this paper derives PAC-Bayes bounds for the posterior distribution conditioned on samples drawn from the data distribution. They are novel from the perspective of statistical learning theory, verified by the comparison with existing work on the PAC-Bayes bound for conditional distributions (Rivasplata et al. (2020)) and generalization bound for VAE (Chakrabarty and Das, 2021 and Cherief-Abdellatif et al., 2022).
- Many generalization bounds for VAE are systematically derived from a single PAC-Bayes bound, showing its generality.
- The paper is well-written. Both the organization and mathematical descriptions of the paper are appropriate. I had no significant difficulties in reading the paper.

**Weaknesses:**

- The obtained bounds are not uniform with respect to decoder parameters. This restriction affects the sample size rate of the bounds: If I understand correctly, the O(1/n) terms in the upper bounds come from the fact that the decoder is a single hypothesis so that the (non-uniform) Hoeffding bound can be applied.
- The decoder outputs mean parameters only, and the variance of the distribution modeled by the decoder is fixed. This architecture is different from the one we often use practically.

**Questions:**

By applying uniform concentration inequality with respect to the decoder parameters (more specifically, the family of loss functions parametrized by the decoder), can we give uniform bound with respect to the decoder parameters at the cost of worsening the sample size rate?

l.134: iid -> i.i.d.

**Limitations:**

This paper discusses limitations in Section 6: (1) the parameters of the decoder are fixed in the derived bounds, and (2) the L1 loss is used as the reconstruction loss instead of the commonly used L2 loss (this is equivalent to modeling the decoder as the Laplace distribution instead of the Gaussian distribution).

Another limitation is that the VAEs used in this analysis have a fixed decoder variance (l.116). If I do not miss any information, it is not discussed whether we can extend the obtained bounds to VAEs that also estimate variance.

---

> ### Author Rebuttal · Authors · 2023-08-06
>
> First, we thank the reviewer for their thoughtful review and insightful comments.
>
> > By applying uniform concentration inequality with respect to the decoder parameters (more specifically, the family of loss functions parametrized by the decoder), can we give uniform bound with respect to the decoder parameters at the cost of worsening the sample size rate?
>
> It is possible to obtain a uniform bound with respect to the decoder's parameters $\\theta \\in \\Theta$, if we assume $\\Theta$ is finite. In this case, the union bound leads to a penalty of $\\log \\lvert \\Theta \\rvert$. The problem with this is that since $\\Theta$ is a set of neural network parameters, this assumption may not be accurate unless $\\lvert \\Theta \\rvert$ is very large, which may significantly worsen the bound.
>
> Regarding uniform concentration inequalities, we believe their usage would require some assumption on the complexity of $\\Theta$, and we do not believe the problem to be straightforward. We agree with the reviewer that obtaining uniform bounds w.r.t. $\\theta$ is important, and we plan on exploring that in future works.
>
> We also mention that the non-uniformity w.r.t. $\\theta$ is due to the fact that in general, PAC-Bayes considers a single loss function, and in this case, the loss function depends on the decoder's parameters.
>
> > the VAEs used in this analysis have a fixed decoder variance (l.116). If I do not miss any information, it is not discussed whether we can extend the obtained bounds to VAEs that also estimate variance.
>
> Indeed, the variance of the decoder is fixed, and there is a way to extend the results to optimize the variance as well. Assuming the standard deviation $\\sigma$ is constant, our bounds yield $\\sigma \\propto \\frac{n}{\\lambda}$ (because in our expressions, the sum of KL divergences is only divided by $\\lambda$, whereas it is usually divided by $n$ as well). Hence, optimizing the decoder's variance $\\sigma^2$ is equivalent to optimizing the hyperparameter $\\lambda$, which can be done for PAC-Bayes bounds, but at a cost.
>
> Most PAC-Bayes bounds (including ours) do not directly allow one to optimize $\\lambda$ (see Section 2.1.4 of Alquier, 2021 and references therein). And, although there are some ways around this restriction, we are not aware of any results that allow one to optimize $\\lambda$ in the general case (meaning continuous $\\lambda$ and unbounded loss). If the loss function is $[0, 1]$-bounded, [1] developed a PAC-Bayes bound uniformly valid for a trade-off parameter $\\lambda' \\in (0, 2)$. For unbounded losses, if one assumes $\\lambda \\in \\Lambda$, where $\\Lambda$ is finite, a union bound argument allows one to make the bound uniform with respect to $\\lambda \\in \\Lambda$, at the cost of $\\log \\lvert \\Lambda \\rvert$. One can still optimize with respect to a continuous set $\\Lambda$, by replacing $\\lambda$ with $\\lfloor \\lambda \\rfloor$. So, for instance, if $\\Lambda = [1, n]$, and we replace $\\lambda$  with $\\lfloor \\lambda \\rfloor$, the penalty is $\\log n$.
>
> In summary, we can use the union bound to extend our results to VAEs that estimate the variance of the likelihood. This extension comes with a penalty depending on the size of the chosen space. We will mention this in the main paper, and add some formal details in the supplementary material.
>
> [1] A strongly quasiconvex PAC-Bayesian bound. Thiemann, N. and Igel, C. and Wintenberger, O. and Seldin, Y.; International Conference on Algorithmic Learning Theory 2017
>
> Once again, we thank the reviewer for their hard work and interesting questions. We hope that the reviewer finds our answers satisfactory, and we will happily answer any further questions.

---

> > ### Comment · Reviewer_sohu · 2023-08-14
> >
> > Thank you for your response. I answer each question in the following responses.
> >
> > **Uniformity w.r.t. decoder parameters**
> >
> > Thank you, I understand and agree with the authors' comments.
> >
> >
> > **Estimation of Variances**
> >
> > I appreciate the authors' explanations. However, it seems what the authors assumed differently from what I have in mind.
> > The authors appeared to discuss extending the theory to the situation where we estimate the variance parameter $\sigma$ **independent of the instance** $\boldsymbol{x}$. That is, the loss function (6) is changed to $\sigma^{\theta}\_{rec}(\boldsymbol{z}, \boldsymbol{x}) = \sigma^{-1}\|g\_\theta(\boldsymbol{z}) - \boldsymbol{x}\|$ (correct me if I was wrong.) On the other hand, I intended the case where the variance parameter depends on the instance $\boldsymbol{x}$: The output of the decoder is $(g\_\theta(\boldsymbol{z}), \sigma_\theta(\boldsymbol{z}))$, and the loss function is (for example)
> > $\ell^{\theta}\_{rec}(\boldsymbol{z}, \boldsymbol{x}) = \sigma\_\theta(\boldsymbol{z})^{-1}\|g\_{\theta}(\boldsymbol{z}) - \boldsymbol{x}\|$.
> > I am sorry for my lack of explanation.

---

> > > ### Author Response · Authors · 2023-08-15
> > >
> > > We thank the reviewer for the clarification. Indeed, we discussed the case when the variance is learned from the training set, but is independent of individual instances $\\mathbf{x}$. Let us define the loss function, as the reviewer suggested, as $\\ell\_{\\text{rec}} =\\frac{1}{\\sigma\_\\theta(\\mathbf{z})} \\lVert \\mathbf{x} - g\_\\theta(\\mathbf{z})   \\rVert$. First, because of the division by $\\sigma\_\\theta(\\mathbf{z})$, we assume there is $\\sigma\_1 >0$ such that for any $\\mathbf{z} \\in \\mathcal{Z}$, $\\sigma\_\\theta(\\mathbf{z}) \\geq \\sigma\_1$. There are two main problems: making sure Assumption 1 is satisfied, and bounding the exponential moment of Theorem 3.1.
> > >
> > > The first problem is equivalent to showing that Proposition 4.1 can be extended to this loss function. Following the second part of the proof of Proposition 4.1 (line 80 in the supplementary material), we need to show that $\\ell\_{\\text{rec}}$ is Lipschitz-continuous. The problem here is that in general, the product of real-valued Lipschitz functions is not Lipschitz. Hence, even assuming that $\\sigma\_\\theta$ is $K\_\\sigma$ Lipschitz (for some $K\_\\sigma > 0$), it does not seem possible to achieve this without additional assumptions. If we assume, in addition, that $\\lVert   \\mathbf{x} - g\_\\theta(\\mathbf{z})   \\rVert \\leq M$ is bounded, then we obtain
> > >
> > > \$ \\ell\_{\\text{rec}}(\\mathbf{z\_1, \\mathbf{x}}) - \\ell\_{\\text{rec}}(\\mathbf{z\_2, \\mathbf{x}}) \\leq
> > > \\left( \\frac{K\_\\sigma M}{\\sigma\_1^2} + \\frac{K\_\\theta}{\\sigma\_1} \\right) \\lVert   \\mathbf{z}\_2 - \\mathbf{z}\_2   \rVert \$
> > >
> > > which implies that Assumption 1 is satisfied with the constant $K\_\\phi \\left( \\frac{K\_\\sigma M}{\\sigma\_1^2} + \\frac{K\_\\theta}{\\sigma\_1} \\right)$, instead of $K\_\\phi K\_\\theta$, and with the family $\\mathcal{E}$ being the set of functions from $\\mathcal{Z}$ to $\\mathbb{R}$ with Lipschitz norm at most $\\left( \\frac{K\_\\sigma M}{\\sigma\_1^2} + \\frac{K\_\\theta}{\\sigma\_1} \\right)$.
> > >
> > > In this case, when the instance space is bounded, the upper bound on the exponential moment (in the proof of Theorem 4.3) is:
> > >
> > > \$
> > > \\frac{\\lambda^2 \\Delta^2}{8n \\sigma\_1^2}, \\quad \\text{ instead of } \\quad  \\frac{\\lambda^2 \\Delta^2}{8n}.
> > > \$
> > >
> > > And under the manifold assumption, we get the following upper bound on the exponential moment (in the proof of Theorem 4.4):
> > >
> > > \$
> > > \\frac{\\lambda^2 K\_*^2}{2n \\sigma\_1^2}, \\quad \\text{ instead of } \\quad \\frac{\\lambda^2 K\_*^{2}}{2n}.
> > > \$
> > >
> > > Note that although the upper bounds on the average distance remain unchanged, the coefficient $K\_\\phi K\_\\theta$ is replaced by $K\_\\phi \\left( \\frac{K\_\\sigma M}{\\sigma\_1^2} + \\frac{K\_\\theta}{\\sigma\_1} \\right)$, which can be larger, specially if $\\sigma\_1$ is very small.
> > >
> > > So in summary, it is possible to extend our results to make the variance of the likelihood dependent on individual samples, but it comes with additional assumptions, making Theorem 4.2 less general. Although it may be possible to improve the results we presented above, perhaps by making different assumptions on the function $\\sigma\_\\theta$.
> > >
> > > As for the results of Section 5, our approach almost "forces" one to consider the loss function we defined in the paper. Indeed, since by default the VAE's generative model only considers the mean $g\_\\theta(\\mathbf{z})$ of the distribution induced by $\\mathbf{z}$ in $\\mathcal{X}$, the approach we took, (Lemma D1) leads directly to lemma B1, which considers a decoder with constant variance. Therefore, although the upper bounds in Section 5 can be affected by $\\sigma\_\\theta$, because of its effect on the empirical losses, the results of Section 5 remain the same. That is, unless one considers a generative model different from the usual $g\_\\theta \\sharp p(\\mathbf{z})$.
> > >
> > > We hope this answers the reviewer's question, and we are happy to answer any further questions. We sincerely thank the reviewer for raising interesting questions.

---

> > > > ### Comment · Reviewer_sohu · 2023-08-16
> > > >
> > > > I thank the authors for the thoughtful discussion in response to my questions. I am mostly satisfied with the authors' answers.
> > > >
> > > > I think the assumption of the lower bound $\sigma_1$ is not strange because we sometimes clamp $\sigma_{\theta}$ from below for numerical stability. On the other hand, since the clamp value is practically small (e.g., $\sigma_1 = 10^{-8}$), derived Lipschitz constants tend to be large, leading to looser bounds.
> > > >
> > > > Regarding Section 5, neglecting the uncertainty of distributions of the encoder and decoder at the inference time is one of the possible choices employed practically. In that sence, $g_\theta(\mu_\phi(\boldsymbol{x}))$ looks more natural to me. However, I think $g_\theta\sharp q_\phi(\boldsymbol{z}\mid \boldsymbol{x})$ is not so a strange choice either, especially for theoretical analysis.

---

> > > > > ### Author Response · Authors · 2023-08-17
> > > > >
> > > > > We thank the reviewer for their thoughtful comments.
> > > > >
> > > > > Regarding the lower bound $\\sigma\_1$, we agree with the reviewer's comments. Furthermore, we would like to mention that the loss of precision due to $\\sigma\_1$ only occurs when one wants to upper bound the expression $\\mathbb{E}\_{\\mathbf{x} \\sim \\mu}\\left[ \\mathbb{E}\_{q\_\\phi(\\mathbf{z} | \\mathbf{x})} \\left[ \\frac{1}{\\sigma\_\\theta(\\mathbf{z})} \\lVert \\mathbf{x} - g\_\\theta(\\mathbf{z}) \\rVert \\right] \\right]$. As mentioned by the reviewer, the uncertainty of the likelihood can be ignored at test time. Therefore, one can still train the VAE using the reconstruction loss defined above (with $\\sigma\_\\theta$), then compute the bounds without the likelihood's variance. This way, one gets an upper bound on $\\mathbb{E}\_{\\mathbf{x} \\sim \\mu}\\left[ \\mathbb{E}\_{q\_\\phi(\\mathbf{z} | \\mathbf{x})} \\left[ \\lVert \\mathbf{x} - g\_\\theta(\\mathbf{z})\\rVert \\right] \\right]$, without the loss of precision due to the size of $\\sigma\_1$. Although as mentioned in the paper, if the optimization objective is not derived from the right-hand side of the bound, then the optimal encoder and decoder minimizing the bound may differ from the ones obtained after training.
> > > > >
> > > > >
> > > > > Regarding Section 5, let us provide some intuition as to why we chose the distributions $g\_\\theta \\sharp q\_\\phi(\\mathbf{z} | \\mathbf{x}\_i)$, for $1 \\leq i \\leq n$. The main objective of Section 5 was to provide the upper bounds on the performance of the VAE's generative model, meaning the distance between $\\mu$ and $g\_\\theta \\sharp p(\\mathbf{z})$. As the reviewer knows, although VAEs are generative models, their optimization objective does not directly provide insights on the generative model's performance (unlike GANs, for instance). So our intuition was this: on one hand, if $\\mathbf{z} \\sim q\_\\phi(\\mathbf{z} | \\mathbf{x}\_i)$ for $1\\leq i \\leq n$, then the reconstruction loss compels $g\_\\theta(\\mathbf{z})$ to look like something sampled from $\\mu$, since $g\_\\theta(\\mathbf{z})$ has to be a decent reconstruction of $\\mathbf{x}\_i$. On the other hand, the KL loss causes all distributions $q\_\\phi(\\mathbf{z} | \\mathbf{x}\_i)$ to be close to the prior $p(\\mathbf{z})$. Thus, $g\_\\theta \\sharp p(\\mathbf{z})$ is close to $\\mu$ if $g\_\\theta \\sharp  p(\\mathbf{z})$ is close to $g\_\\theta \\sharp  q\_\\phi(\\mathbf{z} | \\mathbf{x}\_i)$, for each $1\\leq i \\leq n$ (KL loss), and $ g\_\\theta \\sharp  q\_\\phi(\\mathbf{z} | \\mathbf{x}\_i) $ is close to $\\mu$ (reconstruction loss). Therefore, in order to upper bound $W\_1(\\mu, g\_\\theta \\sharp p(\\mathbf{z}))$, we found it natural to start by upper bounding $W\_1(\\mu, \\hat{\\mu}\_{\\phi, \\theta})$ and $W\_1(\\hat{\\mu}\_{\\phi, \\theta}, g\_\\theta \\sharp p(\\mathbf{z}))$, then use the triangle inequality to obtain an upper bound on $W\_1(\\mu, g\_\\theta \\sharp p(\\mathbf{z}))$. Recall the definition of $\\hat{\\mu}\_{\\phi, \\theta} = \\frac{1}{n} \\sum\_{i=1}^{n}{g\_\\theta \\sharp q\_\\phi(\\mathbf{z} | \\mathbf{x}\_i)}$.
> > > > >
> > > > >
> > > > > The reviewer also mentioned the possible use of the $g\_\\theta(\\mu\_\\phi(\\mathbf{x}))$ in Section 5. Note that if we replace $g\_\\theta \\sharp q\_\\phi(\\mathbf{z} | \\mathbf{x}\_i)$ with $g\_\\theta(\\mu\_\\phi(\\mathbf{x}\_i))$ in the definition of $\\hat{\\mu}\_{\\phi, \\theta}$, then we obtain a distribution on $\\mathcal{X}$ that is only supported on a finite number of samples. On the other hand, if instead of $g\_\\theta(\\mu\_\\phi(\\mathbf{x}\_i))$, we considered $g\_\\theta(\\mu\_\\phi(\\mathbf{x}))$ and took the expectation w.r.t. $\\mathbf{x} \\sim \\mu$, then we would end up with an expression on the right-hand side that is impossible to compute, because $\\mu$ is unknown in general. Therefore, even though we agree with the reviewer about the practical usage of $g\_\\theta(\\mu\_\\phi(\\mathbf{x}))$, it might not be equally adapted to our theoretical analysis.
> > > > >
> > > > > Once again, we are very grateful to the reviewer for engaging in this discussion and raising such interesting questions. We will expand the supplementary material to include our answers to the reviewer's questions, as well as the technical details. We hope the reviewer finds our answers satisfactory, and will happily answer any additional questions.

---

> > > > > > ### Comment · Reviewer_sohu · 2023-08-20
> > > > > >
> > > > > > I thank the authors for further discussions. I am satisfied with the authors' answers.
> > > > > >
> > > > > > **Lower bound of $\sigma_\theta$**
> > > > > >
> > > > > > I understand the change in the Lipschiz constant introduced by making $\sigma_\theta$ instance-dependent does not occur when we consider $\mathbb{E}_{q_\phi(\boldsymbol{z} \mid \boldsymbol{x})} \|\boldsymbol{x} - g_\theta(\boldsymbol{z})\|$.
> > > > > >
> > > > > >
> > > > > > **Choice of Generative Distribution**
> > > > > >
> > > > > > I understand that $g_\theta\sharp q_\phi(\boldsymbol{z}\mid \boldsymbol{x})$ serves as an intermediate scaffold for bounding the difference between the true data distribution and the generative distribution $g_\theta\sharp p(\boldsymbol{z})$ (Theorems 5.2, and 5.4), although it is an interesting quantity of its own (Theorem 5.1, 5.3)
> > > > > >
> > > > > > Again, I thank the authors for answering my questions. I enjoyed the discussions.

---

### Author Rebuttal · Authors · 2023-08-06

We are extremely grateful to all the reviewers for taking the time to read our work and make thoughtful comments and suggestions.

All the reviewers seem to agree that the subject of this work (extending PAC-Bayes theory to conditional posteriors and deriving statistical guarantees for VAEs) is important, and the theoretical results are novel and significant. The reviewers also seem to agree that the paper is well-written and the assumptions and results are clearly presented.

Some of the reviewers inquired about the numerical behavior of our bounds. We agree that this is an interesting question, and we are working on the implementation and some experiments on toy datasets. We will add the results to the final version of the manuscript, and the code will be publicly available. Nevertheless, we emphasize that our primary objective was to develop novel theoretical results and present them with as much clarity as possible. Given the importance of VAEs in the Machine Learning community and the lack of statistical guarantees, our goal was to establish a new theoretical framework for the analysis of the statistical properties of VAEs. We hope our work will foster new ideas and insights in the community.

Once again, we thank the reviewers for their hard work, and positive assessment of our manuscript.

---

### Decision · Program_Chairs · 2023-09-21

**Decision:**

Accept (spotlight)

**Comment:**

All reviewers are excited of the novel approach and the new perspective that PAC-Bayesian theory adds to generative learning. We are excited to see this work at NeurIPS